# Comparative meta-analysis of barely transcriptome: Pathogen type determines host preference

**Zahra Soltani**[1], **Ali Moghadam**[1]*, **Mohammadreza Shamekh**[2]

**1** Institute of Biotechnology, Shiraz University, Shiraz, Iran, **2** Department of Horticulture Science, University of Hormozgan, Bandar Abbas, Iran

* ali.moghadam@shirazu.ac.ir

## Abstract

Fungi and aphids show mutual interactions on barley pathogenesis. Fungi promote pathogenesis, while aphids either weaken or strengthen the infection. Otherwise, fungi alter aphid behavior and performance, further highlighting their complex interactions. Characterizing these synergistic and antagonistic interactions is crucial for understanding pathogenesis. Therefore, we performed meta-analysis and co-expression gene network analyses of the barley transcriptome in response to fungus and aphid based on hormone signaling pathways. We selected 13 studies, including 380 fungal infection samples, 48 aphid-attack samples, and 34 hormone-treated samples. We showed that 1.1% of DEGs were common between fungal and aphid-related datasets, while only 0.1% of DEGs were shared among all datasets. In addition, 70% of common DEGs were uniquely regulated by JA or SA signaling. In contrast, 30% of DEGs were regulated by both JA and SA simultaneously. Regulatory element analysis revealed that 85% of DEGs contained at least one binding site from AP2/EREBP or C2H2 zinc-finger factors that show substantial roles in SAR/ISR pathways during plant defense. Gene network analysis identified key hub genes, including *SSI2, PAD2, RPS1, RPS17, SHM1, CYP5,* and *RPL21C*, which influence plant host preference in response to pathogens. Moreover, we identified novel hub genes with unknown functions that potentially interact with the genes involved in defense responses and host preference. This study presents the first systems biology analysis of barley transcriptomic responses to heterotroph/biotroph cross-talk focusing on the preference and performance of *Rhopalosiphum padi*. Our findings suggest critical insights into the molecular mechanisms underlying barley defense responses and identify valuable candidate genes to developing pathogen resistance genotypes in agricultural systems.

## Introduction

Crops are constantly exposed to various pathogens that affect their growth, development, and yield, threatening food security and agricultural productivity worldwide. In addition, the ever-changing climate is constantly alter the growing region of plants through the increased frequency and intensity of these pathogens. Therefore, encountering crops with pathogens, individually or in combination, is likely to become more severe in the future. While research

**Data availability statement:** All relevant data are within the manuscript and its Supporting Information files.

**Funding:** The author(s) received no specific funding for this work.

**Competing interests:** The authors have declared that no competing interests exist.

**Abbreviations:** DEGs, Differentially Expressed Genes; PKT3, Peroxisomal 3-Ketoacyl-Coenzyme A Thiolase; PRs, Pathogenesis-Related genes; SSI2, Suppressor of Salicylic Acid-Insensitive 2; LOX2, Lipoxygenase 2; OPR3, Oxophytodienoate-Reductase 3; AOS, Allene Oxide Synthase; MeJA, Methyl Jasmonic Acid; SA, Salicylic Acid; NPR1, Nonexpressor of Pathogenesis-Related Genes 1; SAR, Systemic Acquired Resistance; ISR, Induced Systemic Resistance; HR, Hypersensitive Reaction; PKs, Protein Kinases; WGCNA, Weighted Gene Co-expression Network Analysis

has advanced our understanding of host-pathogen interactions under controlled conditions [1,2], less attention has been given to the complex cross-talk between multiple pathogens in field environments.

Fungi and insects are considered to be the dominant pathogens responsible for a severe decrease in crop yields, but less attention has been paid to the interaction between these pathogens and the host plant [3]. Fungi are known for their ability to cause > 70% of the significant diseases in crops [4]. Fungal species, such as *Fusarium*, *Botrytis*, and *Magnaporthe* are the most prevalent pathogens of crops [5]. Other destructive invaders are insects and other serious pests that damage crops both directly or indirectly by transmitting pathogens [6]. Aphids, as phloem-feeding insects, release saliva containing proteins and effectors that modulate plant defenses and facilitate infection [7]. Against these attacks, several host genes, such as immune-related proteins or protein inhibitors, are induced.

Barley (*Hordeum vulgare L.*) is the fourth most significant cereal and a genetic model, susceptible to various damaging aphids and fungi. Specific wild barley (such as *Hordeum vulgare* ssp. spontaneum) genotypes that lead to a decrease in aphid and fungal growth have been identified [8]. However, the specific host traits that reduce aphid growth and their induction in response to attacks remain unclear. Although there is little information on the simultaneous interaction of fungi and insects with the host plant, several recent studies have specifically investigated how fungi impact the natural enemies of herbivores. In such tri-trophic systems, variations in plant quality can significantly affect the activity and population dynamics of insect pests (primary consumers), their natural enemies such as parasitoids (secondary consumers), and hyperparasitoids (tertiary consumers). For example, in barley ecosystems, aphids are often controlled by parasitoid wasps, which may themselves be targeted by hyperparasitoids. Understanding these interactions is crucial for developing sustainable pest management strategies, as changes in plant quality can cascade through all trophic levels, ultimately affecting the entire ecosystem.

Although herbivores and fungi often share the same individual hosts [9], there is a lack of comprehensive insight into how fungal infections affect the function and host plant preferences of insect pests. Since the majority of the attention has been concentrated on the direct effects of fungal pathogens on plant health [10], their indirect effects on herbivorous insects have received less investigated.. Plant-associated fungi may indirectly affect the behavior and abundance of herbivorous insects by enhance (e.g., water-soluble carbohydrates, [11]) or reduce (e.g., nitrogen levels, [12]) the availability and intra-plant allocation of nutrients [6,9]. In particular, it has been found that the intensity of insect-negative responses to host fungal infection differs based on insect feeding (chewing, phloem-feeding, and sucking), fungus life history traits (biotrophic, necrotrophic, and endophyte), and the spatial scale of fungus-aphid cross-talks. However, some aspects have remained unclear yet [9].

According to previous studies, both the preference of crop hosts and the function of insect natural enemies, such as biotrophs, necrotrophs, or endophytes, are negatively affected by fungal pathogen infection. Insect pests feeding on fungus-infected hosts can alter plant nutritional value [13], volatiles [14], defensive compounds [15], physical appearance, and potential mycotoxins caused by fungal infection [16]. In hosts colonized by biotrophic infections, both the preference and performance of insects were negatively affected, whereas necrotrophic fungi decreased insect preference over performance, and endophytes decreased only insect pest performance [9]. However, a critical link between hosts and pathogens are established through plant defenses, which require efficient molecular mechanisms to detect multiple signals and respond appropriately, often involving genes like transcription factors (TFs) and hormone-responsive factors [17].

Crops are confronted with a variety of biotrophic and necrotrophic fungi, which trigger salicylic acid (SA) and jasmonic acid (JA) signaling pathways in the immune response. SA is generally activated against biotrophic/hemibiotrophic fungi and sucking insects [18], whereas JA is mainly negatively reacts to necrotrophic pathogens and chewing herbivorous insects [18,19]. The reciprocal contrast between SA and JA pathways can influence insect performance on infected hosts; piercing-sucking insects perform poorly on biotroph-infected plants, whereas chewing insects struggle more with necrotroph-infected plants [18]. These interactions lead to various defenses, which in turn impact multiple groups of plant natural enemies [20].

Large-scale omics data and systems biology approaches, in particular, provided valuable insights into the gene networks underlying host-pathogen interactions [21]. For instance, meta-analysis is a potent strategy that integrates transcriptomic data to recognize core gene sets and regulate their complex traits [22–24]. The bird cherry-oat aphid (*Rhopalosiphum padi*), powdery mildew (biotrophic), and fusarium (hemibiotrophic) are economically important barley enemies. To investigate whether the primary categories of fungal pathogens vary in their indirect impacts on advanced trophic (such as aphids) levels, we conducted a meta-analysis and WGCNA. Moreover, this research discusses the major roles of the cross-talk between fungus-pest and host to illustrate how hormonal signaling is involved in pathogenicity preference. Candidate regulatory elements were identified through functional enrichment analysis of metabolites, TFs, protein kinases (PKs), and microRNA families. This will help us understand how biotrophic and hemibiotrophic pathogens can alter the patterns of cross-talk between pathogens and plant hosts in natural and agricultural ecosystems. A general perception of how fungal pathogen infections may indirectly affect cross-talk between hosts and insect herbivores remains undiscovered, a significant gap that this research begins to address.

## Methods

### Data collecting

The overall framework for data mining and integration of barley transcriptome datasets using systems biology approaches to uncover essential genes associated with fungus-aphid interactions with hormonal cross-talk that affects aphid behavior for host preference is illustrated in Fig 1. The raw transcriptome datasets of barley related to fungal infections, aphid attacks, and hormonal treatments were retrieved from the Gene Expression Omnibus (GEO, www.ncbi.nlm.nih.gov/geo/) and Array Express (www.ebi.ac.uk/arrayexpress) databases (Table 1). Only datasets deserving the Minimal Information about a Microarray Experiment (MIAME) [25], including studies with relatively similar genetic backgrounds without any mutated or transgenic samples, with suitable quality, and biological and technical replicates of controls and treatments were selected. A total of 13 studies, including 380 fungal infection samples, 48 aphid-attack samples, and 34 hormonal treatment samples (JA, GA, and MeJA), were selected for comparative analysis. The raw data files (CELL) of five studies were included in two platforms, Affymetrix and Agilent.

### Data processing

The Affymetrix and Agilent expression arrays are widely used in high-throughput studies. Because the microarray data have shown the levels of noise and biases, the first step for data analysis is pre-processing. The procedure for raw data consists of background correction, normalization, and summarization [37]. Normalization of the Affymetrix raw data in Expression Console software was applied using the robust multichip average (RMA) method. The RMA algorithm removes unspecific background intensities of scanner images and can process many arrays simultaneously with high accuracy and precision [38]. After background correction,

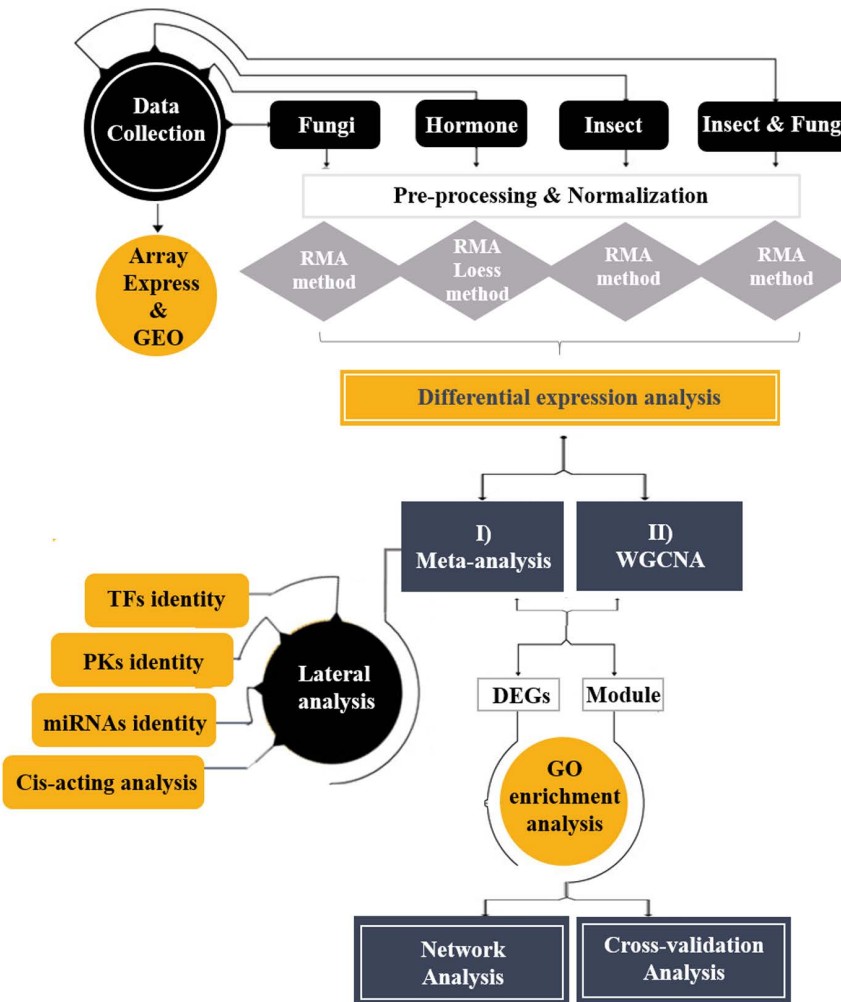

**Fig 1. Schematic overview of applied integrative systems biology approaches on barley transcriptome data.**
Meta-analysis and WGCNA consider the defense responses against fungal pathogens that might affect aphid
behavior for host preference. This insight integrates the hormonal signaling pathways to show specific fungus-aphid
interactions.

quintile normalization was performed and the Median Polish method was used to summa-
rize the probe sets [38]. Additionally, Agilent raw data (one color) were processed in the R
program using the LIMMA and LOESS packages. Agilent transcriptome data (two colors)
processing was conducted in two steps using Flex Array software (Ver 1.6.3). Briefly, back-
ground correction was performed using the Normexp algorithm and the Maximum Likeli-
hood Estimation (MLE) method. Subsequently, normalization between arrays was performed
using the quantile approach. Finally, the gene sequences were retrieved from the NCBI/Batch
Entrez database. Nucleotide sequences of each gene were BLASTed against TAIR using CLC
bio software, and probeset IDs were converted into *Arabidopsis thaliana* orthologs.

## Meta-analysis

The meta-analysis was performed individually using fungal infection, aphid attack, and
hormone treatment datasets and in combination using fungal infection and aphid treatment

**Table 1. Samples were retrieved from GEO and Array Express.**

| Accession | Type | life-style | Platform | Control | Treatment | Reference |
|---|---|---|---|---|---|---|
| E-GEOD-14930 | Powdery Mildew | Biotrophic | GPL1340, Affymetrix | 6 | 30 | [26,27] |
| E-GEOD-33398 | *Fusarium* Head Blight | Hemibiotrophic | GPL1340, Affymetrix | 6 | 6 | [28] |
| E-GEOD-33407 | *Fusarium graminearum* | Hemibiotrophic | GPL1340, Affymetrix | 22 | 22 | [29] |
| E-GEOD-61644 | *Blumeria gramins* f. sp. hordei | Biotrophic | GPL1340, Affymetrix | 30 | 30 | [30] |
| E-GEOD-33396 | Powdery Mildew | Biotrophic | GPL1340, Affymetrix | 36 | 36 | [26] |
| E-GEOD-33392 | Powdery Mildew | Biotrophic | GPL1340, Affymetrix | 72 | 72 | [30] |
| E-GEOD-20279 | *Blumeria gramins* f. sp. hordei | Biotrophic | GPL1340, Affymetrix | 6 | 6 | [31] |
| E-GEOD-12584 | *Rhopalosiphum padi* L. | Interaction by biotrophic/hemibiotrophic pathogens | GPL1340, Affymetrix | 12 | 12 | [32] |
| E-MTAB-5133 | *Rhopalosiphum padi* L. | Interaction by biotrophic/hemibiotrophic pathogens | GPL1340, Affymetrix | 12 | 12 | [33] |
| E-GEOD-5605 | JA | Response to aphid attack and infection by necro-trophic/biotrophic pathogens | GPL1340, Affymetrix | 2 | 2 | [34] |
| E-GEOD-18758 | GA | Response to aphid attack and infection by biotrophic/hemibiotrophic pathogens | GPL1340, Affymetrix | 3 | 3 | [35] |
| E-GEOD-8712 | GA | Response to aphid attack and infection by biotrophic/hemibiotrophic pathogens | GPL1340, Affymetrix | 3 | 3 | [36] |
| E-GEOD-41515 | MeJA | Response to aphid attack and infection by necro-trophic pathogens | GPL15513, Agilent | 9 | 9 | – |

datasets by the MetaDE R package. This package is based on the rankProd method to identify robust up- and down-regulated genes based on the fold change (FC) [39]. To decrease the number of statistical analyses and false errors, 20% of genes exhibiting a bottom expression level with a significant variance of less than 0.2 were removed. Increasing the variance > 0.2 leads to the loss of key genes with significant biological relationships. Conversely, genes with variances < 0.2 are often non-informative with low expression fold change, showing minimal biological relevance and increasing the risk of false positives. Specifically, a variance threshold of 0.2 ensured a focus on genes with significant biological variation, simplified data complexity, reduced computational burden, and improved robustness of the meta-analysis results. The following adjusted t-statistic utilizing 1,000 random shuffles was performed to describe differential expression genes (DEGs), which are genes that show significant changes in expression levels under specific conditions. Finally, up- and down-regulated genes with log2 FC >1 or log2 FC <−1 and adjusted *P-value* (*FDR* ≤ 0.05) were considered statistically significant.

## Co-expression network analysis

We concentrated on the co-expression of genes, which indicates the primary application of correlation network techniques. Co-expression analyses are known for their effectiveness in illustrating mutual relationships between transcripts [40]. A weighted gene co-expression network was created through the R package WGCNA [17] using a united list of normalized expression values of DEGs (*P-value* ≤ 0.05) to detect groups with similar expression patterns. The pairwise Pearson correlation of the DEGs was calculated to establish a correlation coefficient matrix. The correlation matrices were then transformed into adjacency matrices by elevating them to the power (β) that best approximated the scale-free behavior of the resultant networks. Finally, we converted the adjacency matrices into a topological overlap matrix (TOM) to calculate hierarchical clustering. We employed a dynamic tree-cut algorithm to

recognize modules with analogous expression patterns. Next, we approved the step-related modules based on the distinctive interactions between module membership (MM) and the significance gene (GS). The parameters used for each of the datasets included the fungal infection datasets (height cut of 0.3 and minimum cluster size of 30 genes), hormonal treatment datasets (height cut of 0.99 and minimum cluster size of 100 genes), aphid treatment datasets (height cut of 0.5 and minimum cluster size of 50 genes) and the combination of fungal-aphid cross-talk datasets (height cut of 0.3 and minimum cluster size of 30 genes).

### Identification of hub genes

In the following, the genes that have Pearson's coefficient correlation upper than the cut-off value, i.e., ± 0.3 were used for making of biological network by Cytoscape software version 3.6.1 [17]. In other words, the co-expression patterns and interactions of hub genes were exported and visualized using this software. The cytoHubba plugin was used to screen hub genes that exhibited the highest interactions within the biological network. In this way, the computational techniques of Maximal Clique Centrality (MCC) were employed as the most efficient method [17].

### Functional enrichment analysis

To investigate the biological functions associated with the genes or modules identified through meta-analysis and WGCNA, we performed Gene Ontology (GO) enrichment analysis of Biological Process (BP), Cellular Component (CC), and Molecular Function (MF) classifications using the Database for Annotation, Visualization, and Integrated Discovery (DAVID) Web server [41] and g:Profiler [42]. GO enrichment (based on the *P-value* ≤ 0.05 cutoff) was conducted using a distinct list of the TAIR IDs of the DEGs derived from a non-redundant stress file and significant modules for multiple evaluations mapped to each class [17].

### Identification of TFs, TRs, PKs, and miRNAs families

For the identification of transcription factors (TFs), transcriptional regulators (TRs), and protein kinases (PKs), all of the DEGs, were analyzed by iTAK software with default parameters [43]. To extract potentially related miRNAs, a list of mature barley miRNAs was retrieved by CLC-bio software and subsequently analyzed for the targets using the psRNATarget server and standard parameters of hsp size of 20, maximum value expected of 3.0, a length size of 17 bp up- and 13 bp down-stream, and a translation inhibition range of 9–11 nt. The obtained targets were evaluated against the DEGs. Finally, for further functional insight, the identified miRNAs were analyzed with the enriched GO identifier using the Gene Ontology database.

### Cis-acting element analysis

To detect the enriched cis-regulatory motifs within the promoters of the DEGs, 1000 bp upstream of the transcription initiation point of each DEG was retrieved from the Ensembl database (http://plants.ensembl.org). Cis-acting element analysis was performed using the Arabidopsis because about one-third of the barley DEG sequences available at BioMart did not meet annotation and quality standards (i.e., < 800 bp or > 200 N) [44]. Then, all the conserved motifs were retrieved from the MEME website (meme.nbcr.net/meme/intro.html) using an *E-value* of < 10e-4 [45]. In the next step, we aligned the obtained motifs against the JASPAR CORE 2022 website utilizing the Tomtom v 5.0.1 tool (http://meme-suite.org/tools/tomtom) using a *P-value* cut-off of 5e − 3, and then the sequences with significant similarity evaluated with known promoter motifs were preserved. The purpose of this step was to eliminate redundant motifs and determine the known motifs. Finally, we performed an enrichment

analysis of the motifs using the GoMo tool based on Arabidopsis annotation (http://meme-suite.org/tools/gomo) [46].

### Protein-protein interactions (PPI)

PPI networks were developed using protein interaction data obtained from STRING (https://string-db.org/) [47]. The distinct collection of TAIR IDs of each dataset was BLASTed against the STRING database with default settings (highest required interaction score = 0.9). Finally, the networks were visualized using the Cytoscape software, and essential hub genes in the networks were identified based on their degree of connectivity.

### Cross-validation analysis

To validate the findings of the meta-analysis, leave-one-out cross-validation (LOOCV) was applied to raw expression values of hub genes across each of the four datasets derived by supervised WGCNA. In this strategy, the original datasets were divided into training and test series. In this manner, one sample from the original dataset was sequentially removed for testing and the others for training [48]. In addition, we employed model assessment metrics, including the area under the curve (AUC), which were utilized as the assessment criteria for the model. The AUC value was employed as a numerical indicator of model performance, which was categorized as follows: par (0.5–0.6), intermediate (0.6–0.7), well (0.7–0.8), very well (0.8–0.9), and superior (0.9–1).

## Results

### Pre-processing and normalization

Pre-processing using two approaches, RMA and LOESS, identified some noises and eliminated or reduced the effect of existing noises on the machine learning algorithms (S1 Fig). To evaluate estimation bias, we conducted a parametrically designed test in which actual gene expression counts were generated from a lognormal distribution with cell size as a covariate. The actual values of all the involved parameters were adjusted to approximations based on empirical data.

### Meta-analysis and identification of DEGs

We applied two systems biology approaches to multiple transcriptomic experiments in barley to identify specific gene expression changes related to hormonal signaling in response to aphid and fungus invasions. After performing the moderated t-statistic, the Rank Prod technique was applied to examine the transcriptome data and to identify genes with differential expression. Each dataset contained more than 20,000 probeset IDs representing approximately 7,000 genes.

According to the findings, the meta-analysis of hormonal datasets identified 308 significant DEGs (*FDR* < 0.05). Among DEGs, 177 and 127 up- and down-regulated genes were detected respectively (S1 Table). After a meta-analysis of fungal datasets, 1175 up- and 1356 down-regulated genes were extracted (S1 Table), and for aphid datasets, 35 up- and 27 down-regulated genes were introduced (S1 Table). Finally, an integrative meta-analysis of the fungus-aphid datasets identified 346 DEGs (*FDR* < 0.05). Venn diagrams of the total DEGs showed 169 up- and 176 down-regulated genes (S1 Table). Indeed, the results identified genes that were not recognized in individual studies, indicating the extra ability of this approach to determine effective genes.

We reported 263 (9.7%) common genes between fungal and cross-talk fungus-aphid DEGs, 34 (1.2%) common genes between aphid and cross-talk fungus-aphid DEGs, and

30 (1.1%) common genes between fungal and aphid DEGs ([Fig 2a]). These shared DEGs (such as RPS, RPL, NF-YC, NAC, and CYP) are of particular importance because they represent key regulatory nodes that may integrate plant responses to multiple biotic stresses, such as fungal infections and aphid infestations. Hence, the molecular signatures of the shared DEGs suggest that similar functional pathways are modified in response to both pathogens, highlighting the interconnected nature of the plant defense mechanisms. Herbivorous insects and pathogens that inhabit the same host species, cross-talk with one another, either directly or indirectly. Plant interactions with fungal pathogens may affect the general structure of insect populations, leading to alterations in herbivory and intensity via competitive or facilitative mechanisms [49]. These results indicate complex interrelationships between fungi and aphids in barley mediated by shared molecular pathways.

In the following, 119 (4.1%) common genes (such as RPL, CYP, TRA2, and TIFY) between hormonal and fungal datasets, 5 (0.2%) common genes between hormonal and aphid-treated datasets, 14 (2.2%) common genes between hormonal and combined fungus-aphid datasets, and 4 (0.1%) common genes among all datasets (such as PAL and APE) were identified ([Fig 2b]). The existence of common DEGs shows that specific hormonal-responsive genes might have unique roles in the response to pathogens in barley. For instance, genes regulated by the JA/SA signaling pathways may play dual roles in mediating responses to both fungal pathogens and herbivorous insects. As a result, infection by a pathogen or attack by a herbivorous insect can elicit varying defenses, which in turn impacts multiple groups of natural enemies of plants in the ecosystem. These findings underscore the importance of shared DEGs in coordinating plant responses to multiple biotic stresses and provide valuable insights into the molecular basis of plant-pathogen and plant-herbivore interactions.

## Functional enrichment analysis of DEGs

Enrichment analysis was conducted separately on the DEGs resulting from each study to investigate the other potential functions of these genes. These results suggested that *H. vulgar*

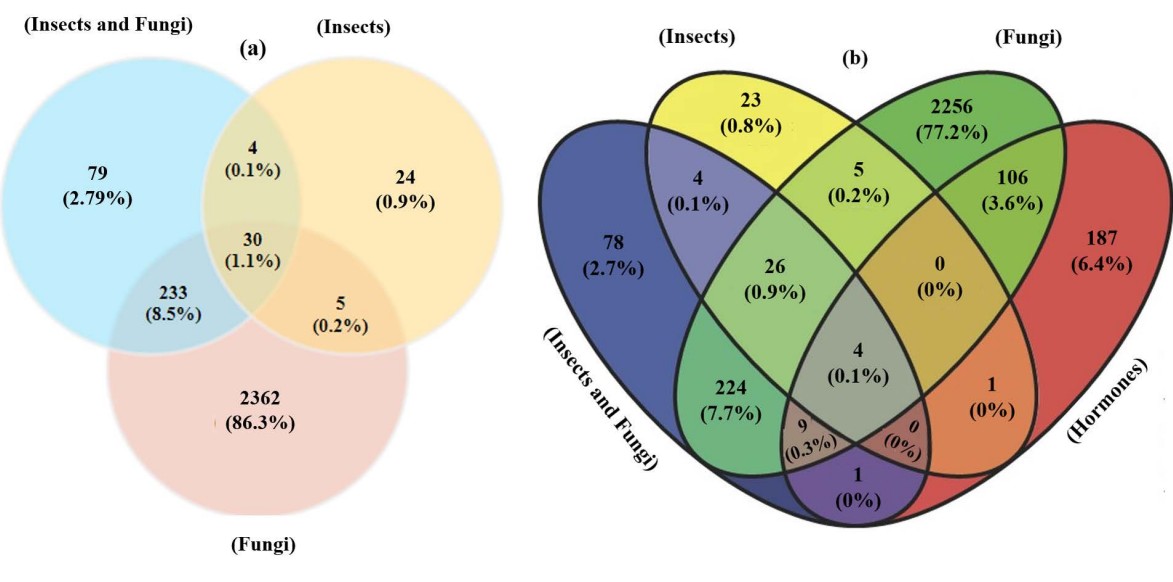

**Fig 2. Venn diagram and comparison of DEGs obtained from individual meta-analyses.** a) Different comparisons between fungal, aphid, and fungus-aphid DEGs. b) Comparison of total DEGs from four separate meta-analyses.

exhibits common and distinct molecular reactions for survival under multiple stress conditions. Heat map charts showing the distribution of up- and down-regulated GO categories in BP, CC, and MF were generated to evaluate the functional similarities and differences across each of the four types of datasets (Fig 3). It is significant to recognize these shared and distinct reactions under fungal infection and aphid and hormone treatments to understand the cross-talk.

Initially, 308 hormonal DEGs were explored for GO and gene network analysis (Fig 3a). These genes were associated with processes of response to JA (6% genes), toxic substances (45% genes), phenylpropanoid biosynthetic process (5% genes), sterol biosynthetic(4% genes), nematodes (9.5% genes), defense responses (10.5% genes such as *PGIP1, LOX3, SOBIR1, AT2G15220*, and *COL1*), oxidative stress (9.5% genes), fungus (5% genes), SA (8% genes), and auxin efflux (7% genes such as *ABCB15, ABCB1*, and *RHM1*). In the class of CC, the most significantly enriched GO terms comprised chloroplast, cytosol, plasma membrane, vacuole, and

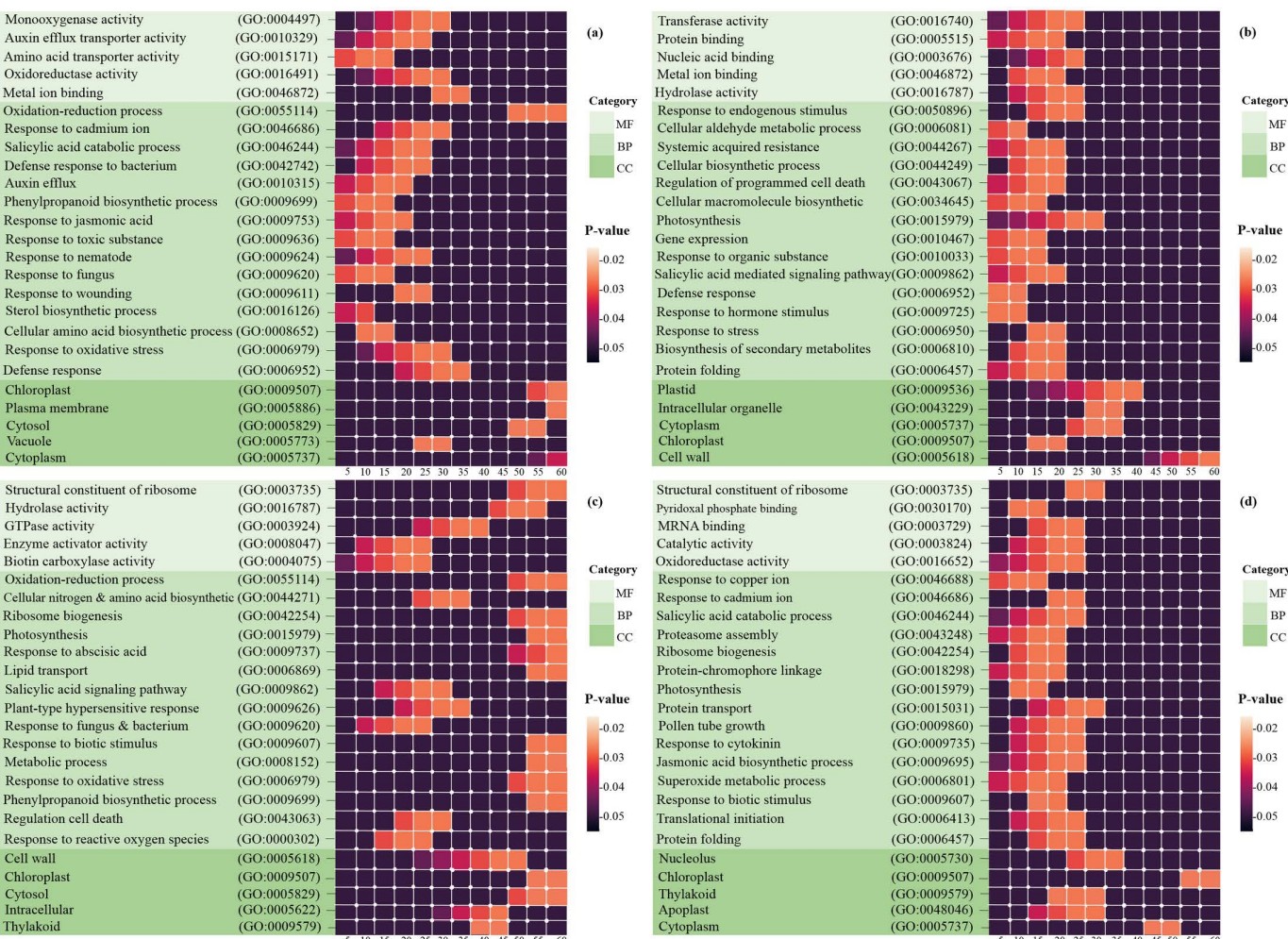

**Fig 3. Binder and heat map plots for enrichment analysis using DAVID.** The significantly enriched GO categories (*P-value* ≤ 0.05) in BP, MF, and CC were associated with immune reactions. The horizontal axis displays the number of genes according to the *P-value*, whereas the vertical axis shows the GO terms in the MF, CC, and BP categories. For completeness and ease of comparison, we defined a set of self-contained plots including a) hormones, b) fungal, c) aphids, and d) fungus-aphid attack cross-talk results, all displayed in the binder plots.

cytoplasm. In the category of MF, amino acid transporter activity, monooxygenase activity, oxidoreductase activity, Auxin efflux transporter activity, and protein binding were observed.

Then the 2531 fungal DEGs were examined. The pathways such as photosynthesis (2% genes), biosynthesis of secondary metabolites (2% genes), oxidation-reduction process (7% genes), JA biosynthetic process (0.6% genes), response to ABA (2% genes such as *ABI1, P5CS1, ABF4*, etc.), response to oxidative stress (2% genes), response to fungus (1% genes), Systemic Acquired Resistance (SAR) (0.2% genes such as *DIR1*), and regulation of programmed cell death (0.01% genes such as *LSD1* and *LCB1*) were strongly expressed in BP. In the MF category, transferase activity, hydrolase activity, and protein/nucleic acid/metal ion binding are prominent. In addition, the CC class included plastid, intracellular organelle, cytoplasm, chloroplast, and cell wall (Fig 3b).

Subsequently, a total of 62 aphid-responsive DEGs were analyzed for GO enrichment. The results of the BP analysis revealed that these DEGs were mainly associated with response to reactive oxygen species (6% genes), oxidative stress (6% genes), biotic stimulus (16% genes), hypersensitive reaction (14% genes), SA-mediated signaling pathway (9% genes), ABA (16% genes), cellular nitrogen, and amino acid biosynthetic processes (16% genes, such as *EMB144* and *MEE32*). In addition, the most abundant GO terms in MF were hydrolase activity, biotin carboxylase activity, and structural constituents of ribosomes, and the most abundant GO terms in CC were the cell wall, chloroplast, and cytosol (Fig 3c).

Finally, GO analysis of the 346 DEGs responsive to fungus-aphid was performed. The processes such as response to ion (7% genes), SA catabolic process (4% genes), proteasome assembly (3% genes), ribosome biogenesis (3% genes), JA biosynthetic process (4% genes), superoxide metabolic process (3% genes), biotic stimulus (3% genes), translational initiation (4% genes), and protein folding (4% genes) were strongly expressed in BP. In the class of CC, the most significantly enriched GO terms were chloroplast, nucleolus, and cytoplasm. In the MF category, oxidoreductase activity, pyridoxal phosphate binding, ribosome structural constituent, mRNA binding, and catalytic activity were observed (Fig 3d). GO enrichment analysis suggested that plants have command and well-established defense mechanisms against pathogens, such as insect pests and fungi. Therefore, it is reasonable that plants contaminated with fungal pathogens may affect insect pests and modify their preferences and functions. While we have tried to explore the diversity and complexity of molecular responses involved in hormonal signaling and pathogens infection through multiple analyses based on the available big microarray data, some undesirable restrictions like the unavailability of more specialized biological data might influence the results. These force us to have a limited range in the list of selected genes based on the optimized filters. Therefore, it would be better to expand these datasets by incorporating newly available transcriptome data. This approach can provide more robust and comprehensive results and resolve existing concerns.

## Common and unique genes responsible for JA/SA signaling

To explore which biological pathways and DEGs responded uniquely or commonly to JA/SA signaling, GO analysis of DEGs was performed using DAVID. Results showed that 18 DEGs, including 12 fungal, 2 hormonal, 2 aphid-related, and 2 fungi-aphid-related datasets in response to the SA signaling pathway and 28 DEGs, including 10 fungal, 15 hormonal, and 3 fungi-aphid-related datasets in response to the JA signaling pathway, were identified. The results revealed that only DEGs related to a specific meta-analysis of aphid datasets did not respond to the JA signaling pathway.

In addition, overlapping DEGs in the four lists (SA up/down-regulated and JA up/down-regulated) were classified. The vast majority (about 70%) of these DEGs were

exclusively regulated by JA or SA signaling, but 30% of these DEGs were regulated by both JA and SA simultaneously. These DEGs, co-induced by SA and JA signals, were expressed in response to wounding, nematodes, fungi, bacteria, hormonal stimuli, reactive oxygen species (ROS), regulator/repressor of transcription, circadian clock, and abiotic stimulus terms. An in-depth analysis of these DEGs uncovered some known or unknown immune regulators, such as *SSI2*, *PTR3*, *AT5G52660*, *AT5G47390*, *GRX480*, and *LHY*. The protein known as the thioredoxin superfamily protein 480 (GRX480) belongs to the glutaredoxin class and plays a crucial role in regulating the redox state of proteins. GRX480 interrelates with TGA factors and inhibits the JA-responsive gene *PDF1.2*. Therefore, *GRX480* is induced by SA and depends on *NPR1*, which might play a role in the interactions between SA/JA cross-talk [50]. The altered lipid levels in the *ssi2* mutants affected the SA- and JA-mediated immune pathways. The *ssi2* mutants showed enhanced resistance against aphids and an antibiosis response in petiole exudates. Interestingly, some JA-induced genes may have a negative impact on the immune response to necrotrophic pathogens and an affirmative impact on biotrophic pathogens [17].

Additional hypotheses arising from the examination of JA/SA co-induced genes included SA biosynthesis-related DEGs (*PAL1*, *DMR6*, *LSD1*, *DIR1*, *WRKY18*, *PNG1*, *HDS*, *PAD4*, *ACD11*, *AHBP-1B*, *GHY*, and *AT4G10490*) and JA biosynthetic genes (*JAZ1*, *NINJA*, *RNS1*, *COL1*, *GAMMA-VP*, *AOS*, *LOX1-2-3*, *STR15*, *OPR3*, *SSI2*, *AIM1*, *OPR2*, *OPCL1*, *ALX5*, *AOC4*, *PKT3*, and *JMT*). *DIR1* is believed to function as a fatty acid transfer protein essential for long-distance signaling in the SAR response. In addition to *AZI1*, it is necessary for the induction of SAR by glycerol-3-phosphate (G3P) and azelaic acid (AA). 13-lipoxygenases (*LOX*), and 12-oxo-phytodieonic acid (*OPDA*), provide precursors for organic compounds biosynthesis, and JA signaling in the defense against insect herbivory [51]. Evidence shows that JA biosynthesis begins with esterification of α-linolenic acid (ALA) in chloroplasts. Next, *OPDA* is produced via a collection of sequential reactions facilitated by *LOX*, *AOS*, and *AOC*. Finally, *OPDA* is transformed into JA-CoA, a precursor of JA, which is facilitated by *OPR3*, *OPCL1*, and *ACX* [19]. Additionally, both *AOS* and *HPL* regulate direct and indirect plant defenses against different Pathogenic agent attacks by impacting the JA and secondary metabolites levels [52]. Therefore, this evidence suggests that certain JA signaling pathway DEGs may improve SA-mediated immunity, and SA could have an affirmative impact on the accumulation of JA.

## WGCNA and module identification

We conducted a co-expression network analysis of the DEGs using the WGCNA package to uncover correlated profiles of gene expression among the studied samples. WGCNA allocates a distinct color label as an identifier for each module. This approach employs mutual correlations of gene expression values to develop co-expression networks that describe modules of strongly intercorrelated genes, which are a mixture of DEGs with known and unknown functions, and identify hub genes that can be applied as biomarkers. Consequently, a soft-thresholding approach with a scale-free model fitting index $R^2 < 0.8$ (Fig 4) was used to enhance the scale-independent topology, while reducing weak correlation and preserving a high average number of interactions. Power values of 10, 12, 14, and 20 for the hormonal, fungal, aphid-related, and fungal-aphid-related datasets were chosen to generate a dendrogram respectively.

Ultimately, the DEGs identified using the dynamic tree-cutting technique were categorized into 7 modules for hormonal datasets containing 141–926 genes per module (Fig 5a, Table 2), 6 modules for fungal datasets containing 473–101 genes per module (Fig 5b, Table 2), 5 modules for aphid-related datasets containing 717–120 number of genes per module (Fig 5c, Table 2),

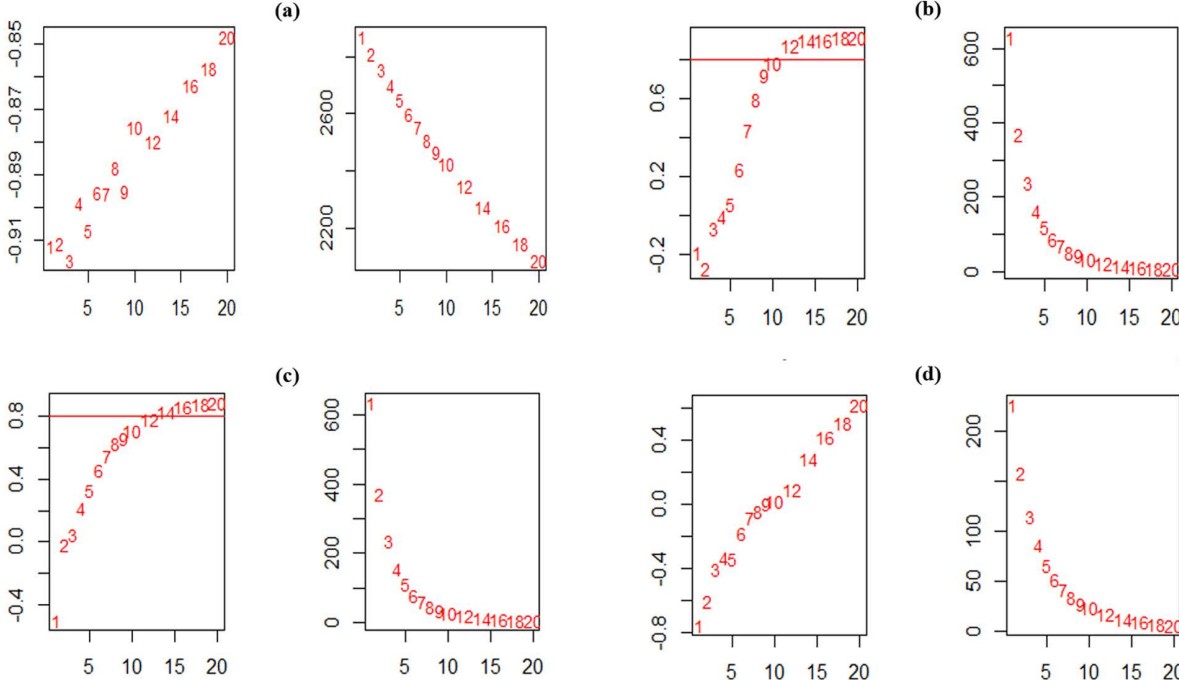

**Fig 4. Identification of soft-thresholding power (β) in WGCNA and module detection for** (a) hormonal datasets (b) fungal datasets (c) aphid-responsive datasets and (d) fungus-aphid-responsive datasets. The left panel illustrates the evaluation of the scale-independent fit index for various soft-thresholding powers (β). The right panel shows an evaluation of the average interactions for various soft-thresholding powers.

and also 5 modules for fungal-aphid responsive datasets containing 97–37 number of genes per module (Fig 5d, Table 2), excluding grey module listing genes that did not significantly co-express with any other group of genes.

Gene co-expression networks (GCNs) consist of genes that have similar patterns and are strongly correlated with each other. Hierarchical clustering analysis showed that we obtained two main clusters, each containing 5 sub-clusters for hormonal datasets [17], 4 sub-clusters for fungal datasets, 3 sub-clusters for aphids, and 3 sub-clusters for fungus-aphid datasets (Fig 6). Based on the multidimensional scaling (MDS) of the hormonal datasets (Fig 7a), the genes in the majority of modules, such as blue, green, and turquoise, and genes in the brown, red, and yellow modules showed analogous expression patterns. In the case of the fungal and aphid-responsive datasets (Fig 7b-c), the genes in the blue, green, and turquoise modules exhibited analogous expression profiles. Finally, in the fungus-aphid-responsive datasets (Fig 7d), genes in the green and brown modules showed analogous expression profiles.

In the adjacency matrix heat map, the variance gradient from blue to yellow indicates the degree of connectivity of genes for various modules from strong to weak, whereas the red points the strongest modules associated with the datasets. In the heatmap related to hormonal, aphid, and fungus-aphid datasets (Fig 6a-c-d), the brown or blue modules, and in the fungal pathogen (Fig 6b), the turquoise or red modules demonstrated the strongest gene-gene connectivity derived from TOM dissimilarity metrics. According to the topological overlap matrix (TOM) illustrated in Fig 8, the dark color indicates minimal overlap, whereas the light color represents greater overlap. Sections of light color across the diagonal are groups of genes (Fig 8).

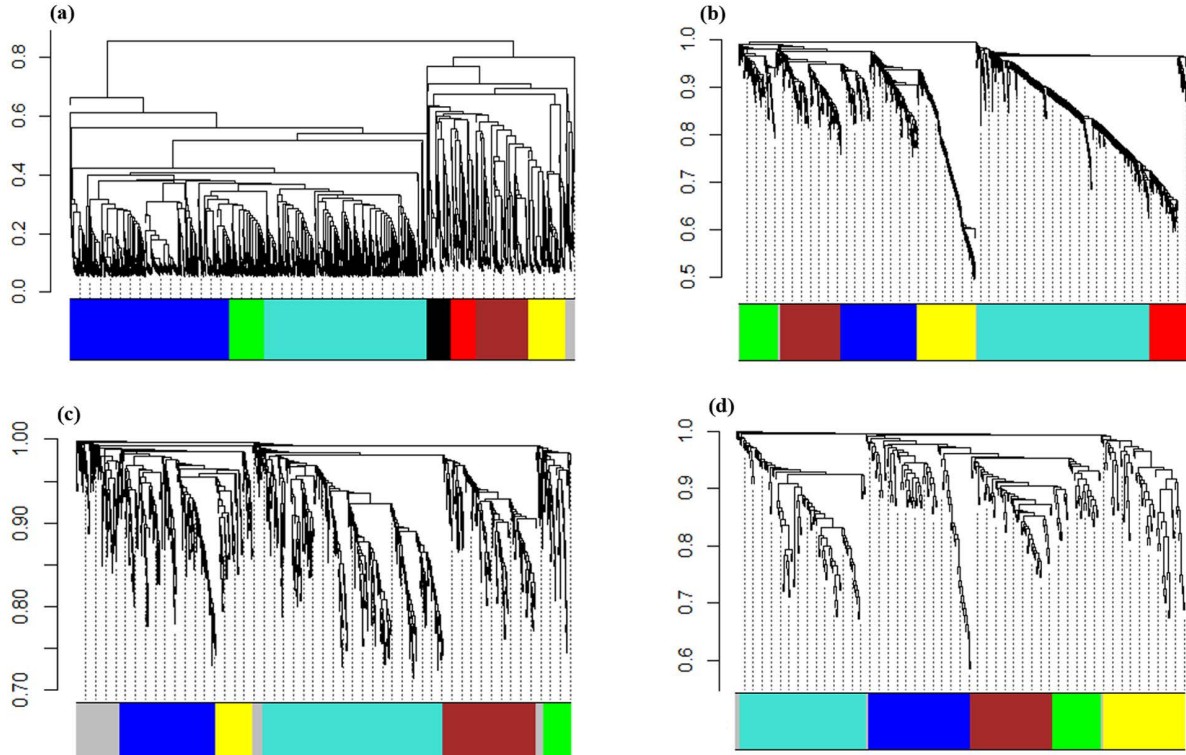

**Fig 5. WGCNA using the dynamic tree cut method.** Cluster dendrogram and modules associated with (a) hormonal datasets (b) fungal datasets (c) aphid-responsive datasets and (d) fungus-aphid-responsive datasets were illustrated. The branches represent modules that consist of strongly intercorrelated gene categories. Each color indicates one distinct co-expression module, while the ends of the branches indicate the individual genes.

In the following, the KEGG pathway annotation of the modules was conducted on the 4 meta-analyses separately (S2 Table). Plants have developed shared molecular strategies to defend themselves against threats from herbivorous insects, fungal pathogens, and hormonal signaling pathways. These include biosynthesis of secondary metabolites, plant-pathogen interactions, biosynthesis of amino acids, photosynthesis, metabolic pathways, carbon metabolism, fructose and mannose metabolism, biosynthesis of antibiotics, and hormone signaling. However, not all strategies are triggered by, nor are they suitable for combating all plant infections. The results of BP showed that SA signaling is frequently activated and effective against biotrophic fungi and sucking insects, necrotrophic fungi, and chewing insects, and mainly stimulates and responds to JA signaling. Finally, we identified a three-way relationship between fungal infection, insects, and the SA/JA hormonal signaling pathways.

### The subject-special and common co-expression profiles under JA/SA regulation

To identify in greater detail the DEG expression profile in response to SA or JA, the analysis focused on specific modules. The genes present within the identified modules may have been co-regulated by JA and SA in concordant or discordant directions (S2 Table). Specifically, in the fungal treatment, the JA/SA co-expressed DEGs such as *AOS, AOC4, OPR2, OPCL1, JAZ1, NAPRT2, NINJA, AT1G29640, ADC2, NTT1, OPCL1, RCD1, ABCG34/DMR6,* and *PAL1* were in the brown module, and the JA co-expressed genes DEGs such as *SSI2, AIM1, PKT,*

**Table 2. Modules of four types of co-expression analysis on each dataset.**

| Total hubs | Total modules | Module name | Datasets |
|---|---|---|---|
| 198 | 926 | Blue | **Hormonal** |
| | 305 | Brown | |
| | 206 | Green | |
| | 142 | Red | |
| | 947 | Turquois | |
| | 221 | Yellow | |
| | 141 | Black | |
| 168 | 208 | Blue | **Fungal** |
| | 167 | Brown | |
| | 104 | Green | |
| | 101 | Red | |
| | 473 | Turquois | |
| | 161 | Yellow | |
| 272 | 442 | Blue | **Aphid-related** |
| | 358 | Brown | |
| | 120 | Green | |
| | 717 | Turquois | |
| | 149 | Yellow | |
| 119 | 79 | Blue | **Fungus-aphid related** |
| | 63 | Brown | |
| | 37 | Green | |
| | 97 | Turquois | |
| | 63 | Yellow | |

*LOX1*, *LOX2*, and *JMT* were in the turquoise and blue modules. *PAL* is the first enzyme in the phenylpropanoid pathway to catalyze the precursors of lignins and phenols, whose activity is tightly regulated by the SA/JA hormonal signaling pathways, which modulate plant defense against biotrophic and necrotrophic pathogens, respectively [17,53,54]. The expression of JA biosynthesis DEGs, such as *LOX2* and *AOC*, suggests that SA does not affect JA biosynthesis [19]. In the brown module, the genes were uniquely up-regulated by SA-enriched functional terms, such as phenylalanine biosynthesis, defense regulation, and immunity suppression. *DMR6* transforms SA into 2, 3-dihydroxybenzoic acid (2, 3-DHBA). It suppresses immunity and negatively regulates defense-associated genes (e.g., *PR1*, *PR2*, and *PR5*) [17]. In other words, they are negative regulators of the defense against pathogens. *JMT* in the blue module shows various co-expression profiles over time after hormonal treatment. These results suggest that *JMT* may catalyze the formation of MeJA from JA. Moreover, its expression was stimulated in response to wounding or MeJA treatment. These results indicated a profile of progressive up-regulation and strongly synchronized co-expression in both MeJA and JA treatments but exhibited a consistent expression pattern and low synchronized co-expression in SA treatment. To recognize in more depth the genes uniquely up-regulated by SA, hormonal datasets were investigated, so that genes such as *LTL1*, *ABCG40*, *IRE1A*, and *OPR1* were identified in the black module. *OPR1* encodes a protein of the α/β barrel fold class of FMN, comprising oxidoreductases, which is induced by SA. This enzyme is unlikely to be involved in JA biosynthesis, as *In vitro* tests have demonstrated minimal function with the naturally occurring OPDA isomer [17].

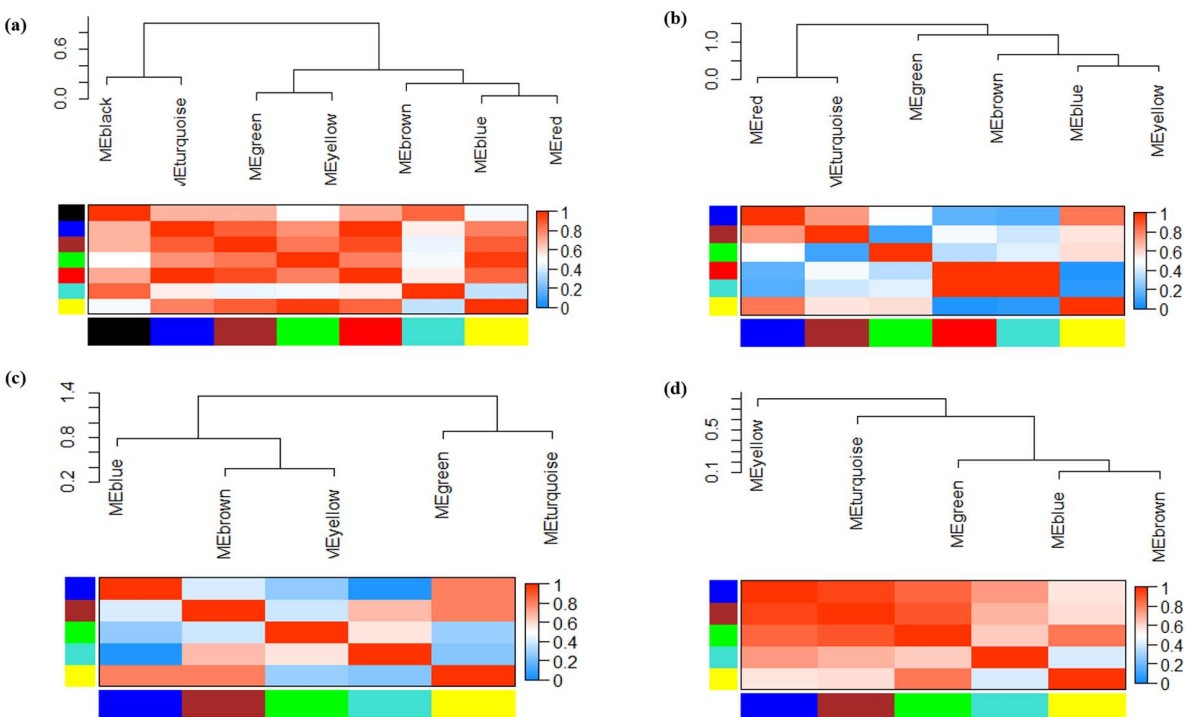

**Fig 6. Module eigengene adjacency was visualized by hierarchical clustering and a heat map.** Each module eigengene encapsulates the DEGs expression pattern related to that class, including (a) hormonal datasets (b) fungal datasets (c) aphid datasets, and (d) fungal-aphid cross-talk datasets. In the heatmap illustrating the relationships between modules, saturated blue, and red colors signify a strong level of co-expression connectivity among modules.

## Screening of hub genes and visualization of gene networks

A network of co-expressed modules was developed to identify the hub genes. After importing the data into Cytoscape and running the CytoHubba application, the top 30 genes evaluated by the MCC calculation method were listed. Finally, we identified 198, 168, 272, and 119 hub genes (S3 Table) in the hormonal, fungal pathogen, aphid-responsive, and fungus-aphid-responsive datasets, respectively. Through GO enrichment, we successfully identified the mechanisms that respond to pathogen infection and hormonal regulation (Fig 9a). Hub genes associated with hormonal modules were highly expressed in the oxidation-reduction processes, pathogen infection responses, signal transduction, phosphate metabolic process, response to pathogens, and hormonal stimulus pathways (Fig 9a). GO findings associated with fungal modules indicated that the major significant BP categories expressed were related to pathways in photosynthesis, ribosome biogenesis, fructose metabolic process, reaction to cadmium ion, cellulose biosynthetic process, oxidation-reduction, gene expression, alcohol metabolic process, cofactor metabolic pathway, resistance to stresses, and defense response. The hub genes associated with aphid modules were highly expressed in protein folding, response to cytokinin, JA biosynthetic process, cellular response to oxidative stress, hydrogen peroxide catabolism, proteolysis, lignin biosynthetic process, response to ABA stimulus [20], and response to oxidative stress. Moreover, within the fungus-aphid modules, the most highly expressed BP categories were related to defense signaling processes (Fig 9a).

Furthermore, using a Venn diagram plot (Fig 9b), we observed intersections of these four datasets and identified significant common hub genes. Among these common DEGs, 28

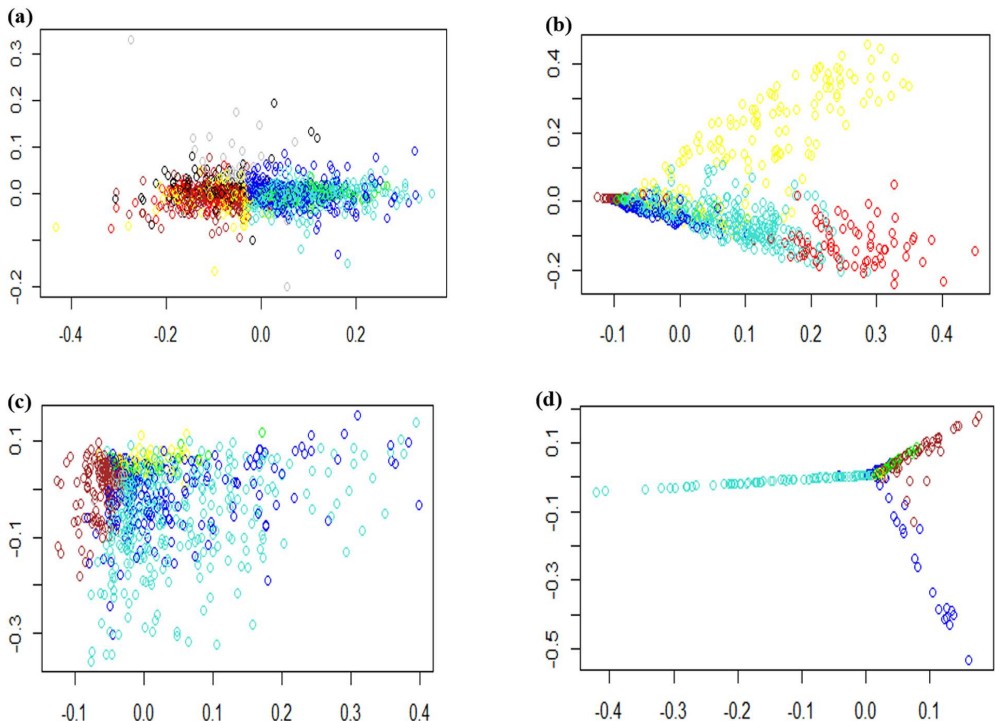

**Fig 7. Multidimensional scaling expansion (MDS) plots of the distinct genes found in each module.** a) hormonal datasets (b) fungal datasets (c) aphid datasets and (d) fungus-aphid cross-talk datasets. The MDS map illustrates the correlation of DEG expression profiles across various modules. The DEGs of various modules are labeled in distinct colors.

important hub DEGs (including 16 down- and 12 up-regulated genes) that interacted with each other were used for GO analysis using the STRING database (Fig 9c). These hub genes were mainly enriched in translation, ribosome biogenesis, defensive response to pathogens, especially aphids, protein folding, photosynthesis, hydrolase activity, SA and JA signaling pathways. Besides, during the studies, strikingly some hub genes with unknown activities such as *AT4G31530*, *AT5G54855*, *AT3G07640*, and *AT1G05720* were identified as promising candidates for subsequent investigation that may be effective on aphid performance and preference in choosing the host plant infected to biotrophic/hemibotrophic pathogens. Finally, to investigate and predict the potential role of these genes, an interaction network with other related genes was constructed using the Genemania algorithm (Fig 9c).

## Identification of TFs and TRs involved in defense responses

Exploring the regulators, including TFs, which mediate interactions among various stress responses is crucial for the strategic modification and resilience of plants facing combined stressors. Precise adjustment of their expression has proven to be a successful approach for the manipulation of complex network pathways in crop development. Therefore, to identify more defense-responsive TFs and TRs and to understand the effect of multiple pathogen cross-talk on hormonal metabolism, we conducted an identification analysis of all DEGs encoding regulatory factors using iTAK software.

In the present study, we identified 7 DEG TFs containing 4 unique TF families expressed specifically under hormonal signaling, among which only the bZIP families were commonly

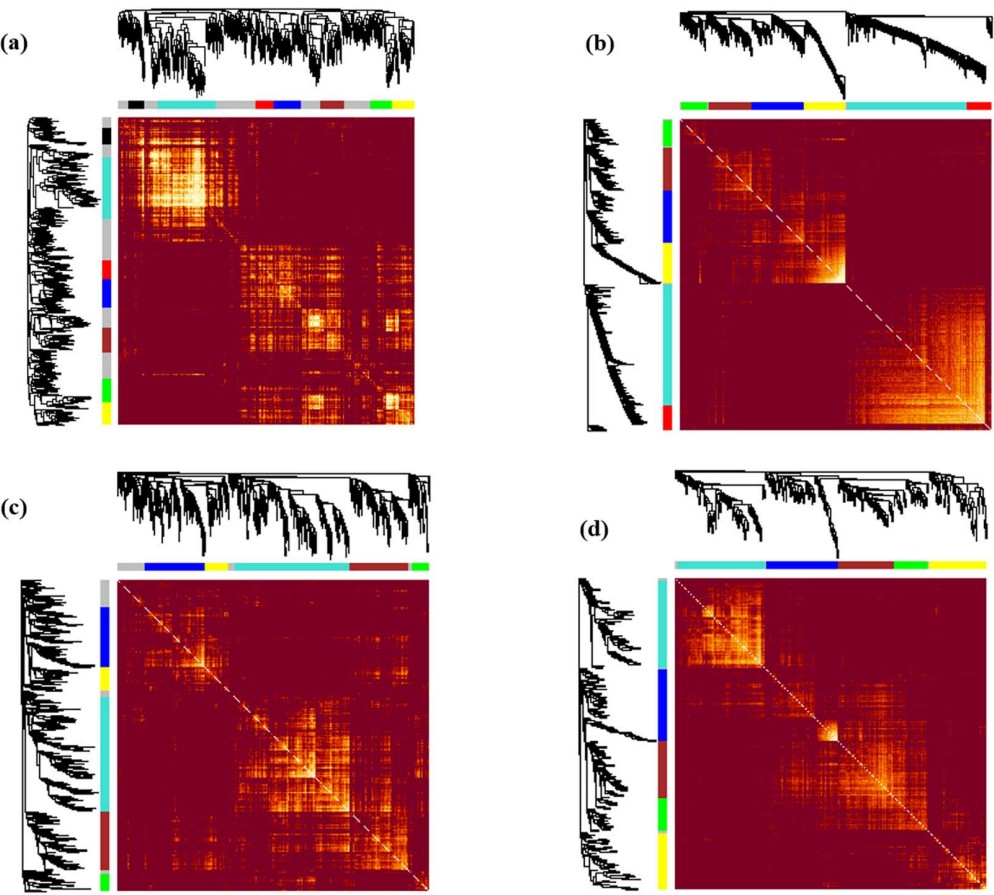

**Fig 8. Co-expression gene network heatmap plots.** WGCNA modules of (a) hormonal DEGs (b) fungal DEGs (c) aphid DEGs and (d) fungus-aphid-responsive DEGs. In the topological overlap matrix (TOM) plot, darker colors indicate minimal overlap, whereas light colors indicate greater overlap across DEGs. Sections with darker colors along the diagonal are associated with groups of genes. Additionally, the corresponding DEGs dendrogram and module are displayed along the left side and top.

down-regulated and most of them showed up-regulation under hormonal signaling conditions (Fig 10a, S4 Table). Among these, C2H2 is the most abundant TF family that directly or indirectly responds to pathogens and environmental stresses.

According to the results, 42 fungal genes encoding TFs were identified and categorized into 18 families MYB, GRAS, and bZIP were the most abundant, and alfin-like, bHLH, C2C2-GATA, CAMTA, GARP-G2-like, NF-YC, SBP, Tify, Trihelix, and zn-clus families showed the minimum number of DEGs (each family had only one gene) (Fig 10a). However, the number and type of up-regulated TFs were less than those of the down-regulated ones. The top 11 up-regulated TFs belonged to the alfin-like, HB-HD-ZIP, MADS-MIKC, CAMTA, Trihelix, and zn-clus families (S4 Table). Among the aphid TFs, it is worth mentioning that only two C2C2-LSD members and one C2C2-GATA gene showed down-regulation (Fig 10a, S4 Table).

Finally, during the investigation of TFs involved in fungus-aphid-responsive cross-talk, a total of 9 TFs were detected. These comprised 6 families, with GRAS and C3H being the most abundant TF families, among which only the MADS-MIK family was up-regulated (Fig 10a, S4 Table). Our results indicated that TFs were strongly affected by the three types of stresses

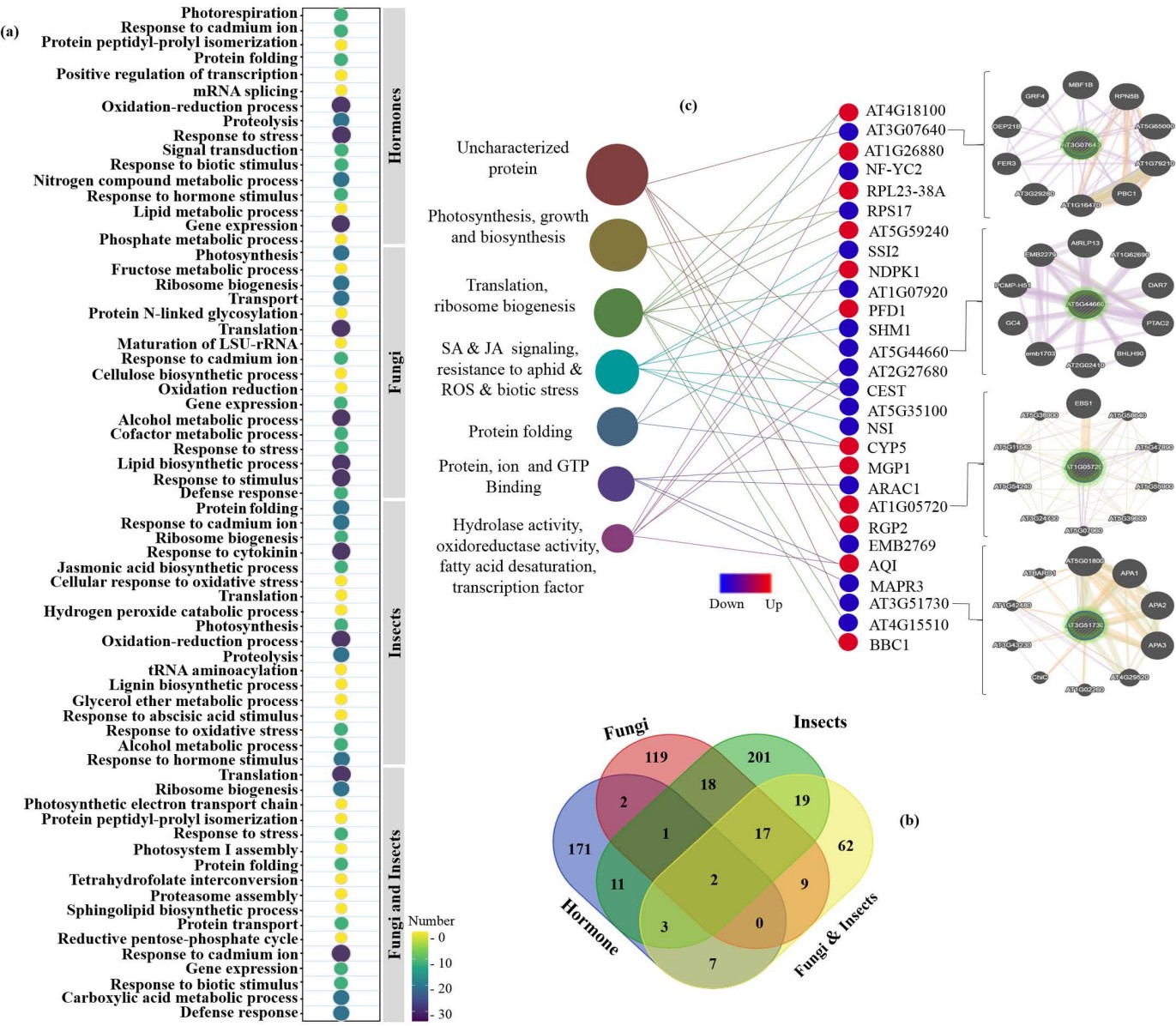

**Fig 9. Gene Ontology (GO) analysis of hub genes.** (a) Allocation of Biological Processes (BP) for the hormonal, fungal, aphid-responsive, and fungus-aphid-responsive hub genes. The GO terms that were significantly associated (*P-value* ≤ 0.05) with the hub gene lists related to each meta-analysis were drawn, along with the number of genes for each GO category. (b) Venn diagram of hub genes obtained from co-expression analysis of each dataset. (c) The 28 significant hub genes (*P-value* ≤ 0.05) were functionally annotated using the STRING database.

and likely played key and crucial functions in multiple stress responses, aligning with their active contributions in both BS and AS resistance.

Given that TRs serve as essential regulators in gene regulatory networks, influencing plant growth, development, and metabolism through the up- and down-regulation of target genes, it would be valuable to identify the genes that encode these proteins and transfer them to crops to enhance their resistance to pathogens. Among the TRs related to hormonal data, it is worth mentioning that only one *GNAT* was down-regulated (Fig 10b, S5 Table). According to the results of fungal datasets, we identified 26 TRs containing 12 unique families, of which

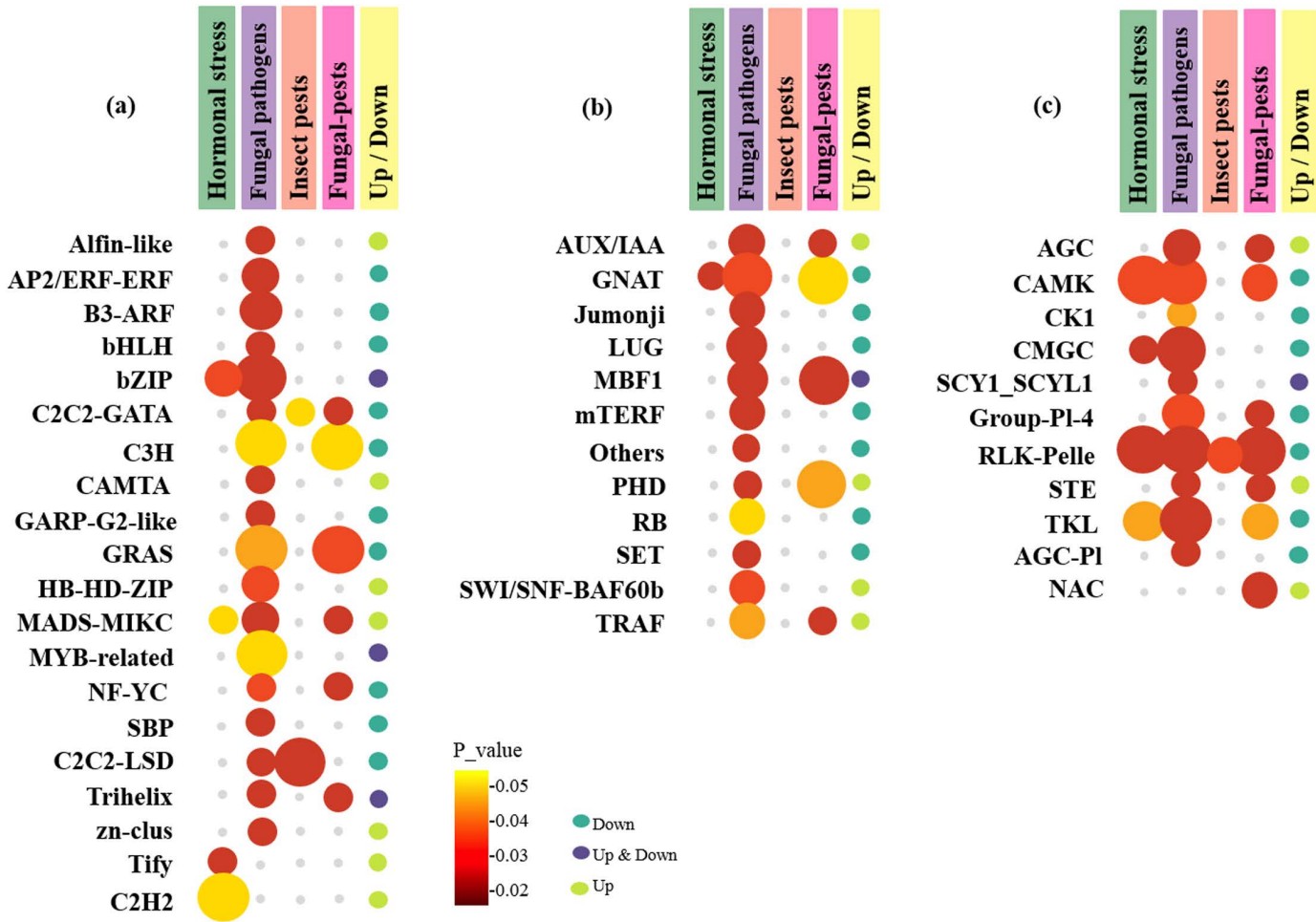

**Fig 10. Bubble heat plot showing further analysis of DEGs between the hormonal, aphid-responsive, and fungus-aphid-responsive datasets.** (a) Distribution of up- and down-regulated TF groups identified in the DEGs. (b) Distribution of up- and down-regulated TR groups identified in the DEGs. (c) Distribution of up- and down-regulated PK groups identified in the DEGs. The bubble sizes represent the number of genes in each dataset. Colors indicant the *P-value* ≤ 0.05. Each column of the bubble plot identifies a dataset and the last column represents the up- and down-regulated genes.

GNAT was the most abundant. Among these, only the GNAT and PHD families were mostly up-regulated (Fig 10b, S5 Table).

No transcriptional regulators were identified among the insect datasets, but in the TRs involved in fungus-insect attack cross-talk, 16 genes from 5 families, including five members of PHD, four members of MBF1, two proteins of AUX/IAA, and four proteins of GNAT, were detected. In total, the number of up-regulated TRs was lower than that of down-regulated TRs, and only two DEGs from the GNAT families were up-regulated (Fig 10b, S5 Table). The present concept may contribute to the understanding of the molecular regulation and signaling interactions between TFs and pathogens. This knowledge could help address challenges and problems in developing potential strategies to enhance the complex regulatory events involved in plant tolerance to pathogen adaptive pathways in barley.

## Identification and classification of PKs

Protein kinases (PKs) play significant roles in signaling pathways involved in the detection of pathogens, phytohormones, and a range of environmental factors. Therefore, it modulates

the activity, function, stability, and localization of other proteins. As shown in Fig 10 and S6 Table, 29 genes and four PK families were identified in the hormonal datasets (Fig 10c). The majority of PK subfamilies belong to the receptor-like kinase (RLK/Pelle) family, of which 12 were down-regulated and 12 were up-regulated. Interestingly, all members of the TKL and CMGC families were up-regulated. In addition, a total of 72 PKs, such as RLK/Pelle, CAMK, CMGC, and TKL, were predicted in 9 families in the fungal datasets (Fig 10c, S6 Table). RLKs represent the most substantial share among the different classes of protein kinases, exhibiting consistently elevated expression levels and frequently being up-regulated. In the insect datasets, two members of one PK family, RLK-Pelle-LRR-II, were identified, all of which were up-regulated.

Finally, during the investigation of PKs involved in fungus-aphid-responsive cross-talk, a total of 32 PKs were identified. These contained 6 families, of which the RLK/Pelle and CAMK were the most abundant families, among which only one case from the CAMK (CAMK-CAMKL-CHK1) family was down-regulated (Fig 10c and S6 Table).

### Prediction of miRNA targets

MicroRNAs (miRNAs) are recognized to regulate and modulate the signaling of many biological processes. In the present study, miRNAs were predicted using a psRNAtarget. Supplementary Table 7 shows the number of miRNAs detected in the DEGs of all meta-analyses. The identified target miRNAs were confirmed using a meta-analysis of *H. vulgare* transcriptome sequencing datasets for large-scale confirmation of the miRNA-target binding interactions.

These statistical analyses resulted in the identification of 861 conserved miRNAs belonging to 57 families in fungal DEGs, 48 miRNAs belonging to 27 miRNA families in aphid DEGs, and 81 miRNAs belonging to 32 families in hormonal DEGs (Fig 11, S7 Table). In addition, we identified 81 potential miRNAs belonging to 57 families in fungus-aphid response that met the requirements for candidate miRNAs in this study (Fig 11, S7 Table). Comparative analysis demonstrated that the largest number of DEmiRNAs reacted to fungal pathogens (358 up-regulated and 503 down-regulated), fungus-aphid responsive datasets (50 up-regulated and 65 down-regulated), hormonal signaling (51 up-regulated and 34 down-regulated), and finally aphid-responsive (39 up-regulated and 9 down-regulated).

After 4 meta-analyses, we found that 14 DEmiRNAs were common among all datasets, with the same expression direction (up or down). The pathogens-pair-wise comparison demonstrated that fungal and fungus-aphid responsive DEGs shared a maximum DEmiRNAs of 49 with conserved expression profiles, fungus-aphid responsive and hormonal datasets shared 37 DEmiRNAs, fungus-aphid and aphid responsive datasets shared 32 DEmiRNAs, and fungal and hormonal datasets 27 DEmiRNAs. We found that shared miRNAs also had a consistent expression profile among the datasets. Several miRNA families are present in only one member. The frequency of different members of the same miRNA family varies drastically. The expression levels of a few miRNAs, such as miR444, miR156, miR159, miR5049, and miR6192, were extraordinarily high in 4 meta-analyses. However, some miRNAs, such as miR5053 showed very low expression levels.

To clarify the putative function of the genes targeted by DEmiRNAs under multiple pathogen infections, pest attacks, or hormone treatments, the target genes of the DEmiR-NAs were enriched using GO analysis (Fig 12). We found that all target DEGs played a role in response to biotic stimulus, amino acid biosynthesis processes, defense response, superoxide metabolism, and carbohydrate derivative/ADP binding. Hormonal signaling pathways were the most significant terms among all datasets. A significant number of the

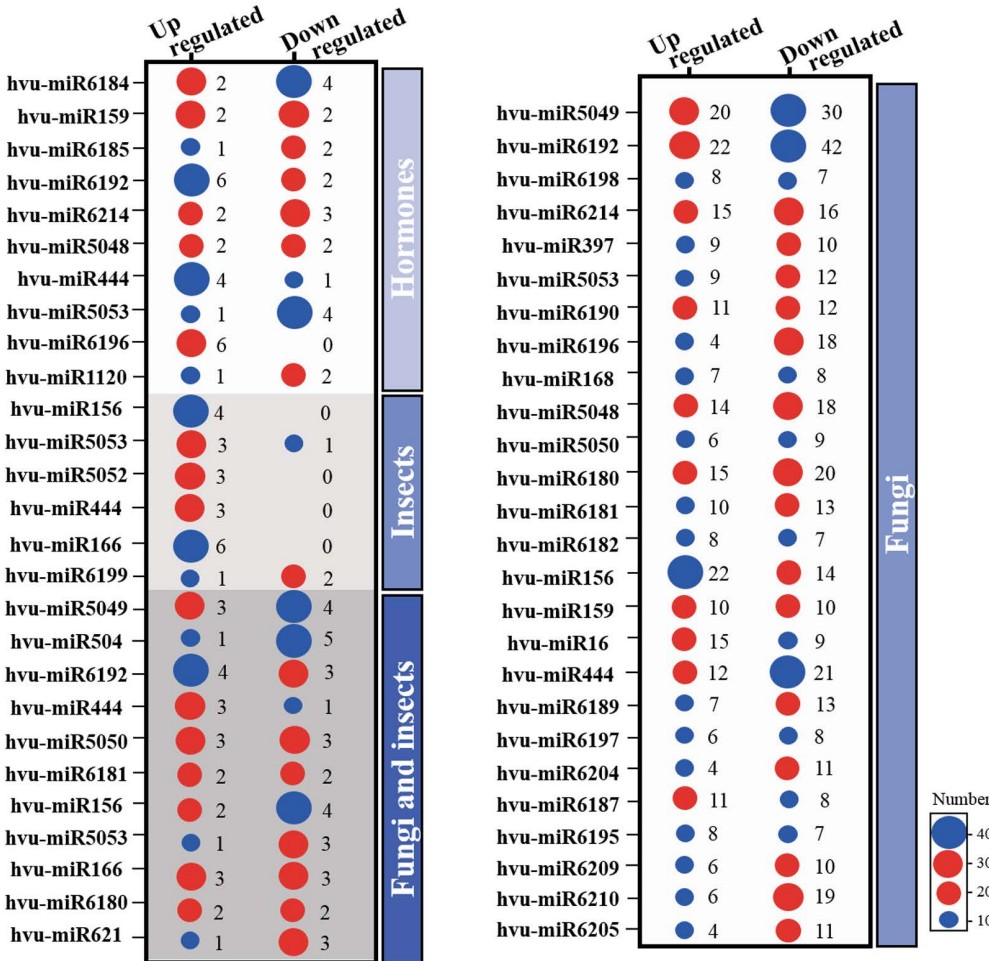

**Fig 11. Bubble plot showing up- and down-regulated differentially expressed miRNAs and their predicted targets in 4 meta-analyses of fungal, hormonal, aphid-responsive, and fungus-aphid-responsive datasets.** miRNA prediction was performed using the psRNAtarget server (http://plantgrn.noble.org/psRNATarget/).

predicted targets were poorly expressed and characterized, suggesting possible new roles for the DEmiRNAs in barley.

Therefore, these miRNAs may form candidates for potential regulatory roles in defense responses and signaling networks.

## Cis-acting elements analysis of DEGs

Examining the cis-regulatory factors within the promoter regions is crucial to understanding gene regulation and function. Promoter region up to 1.5 kbp upstream from the translation start site of each DEG was scanned using Ensemble Plants (http://plants.ensembl.org) tools for the detection of cis-acting elements (CAREs). Arabidopsis provides a standard framework for comparing gene expression at both the genomic and transcriptomic levels, which serves as a comprehensive reference for functional studies [55]. Evidences suggest that despite evolutionary divergence, there are many common regulatory mechanisms, supporting the use of Arabidopsis as a model for understanding gene regulation in barley [56]. Therefore, it can provide insights into gene functions and regulatory mechanisms that may not yet be available for

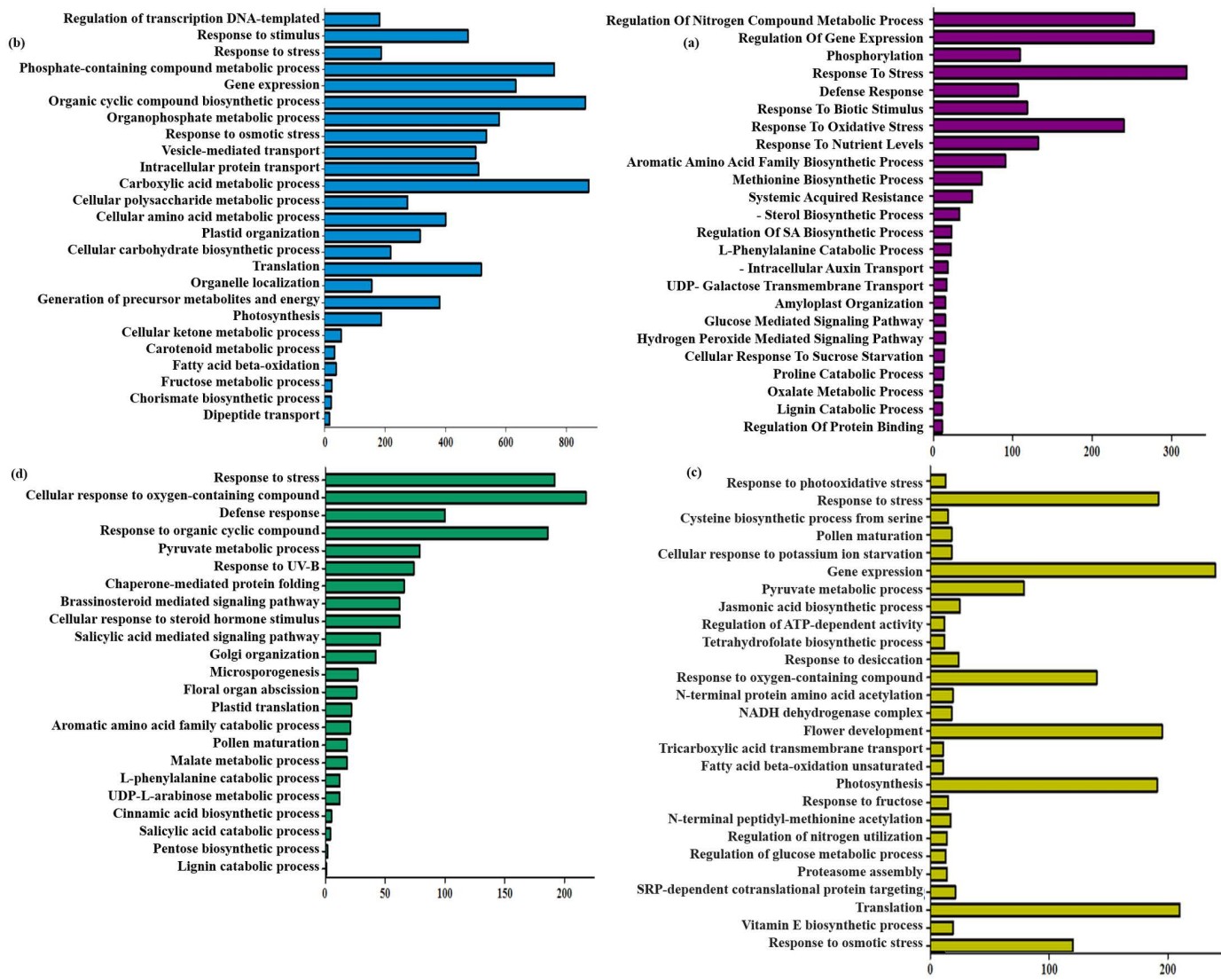

**Fig 12. Functional enrichment of miRNA target DEGs.** Classification of miRNA target DEGs using WEGO tools (http://wego.genomics.org.cn/cgi-bin/wego/index.pl) in 4 meta-analyses: (a) Hormonal target miRNAs. (b) Fungal target miRNAs. (c) Aphid-responsive target miRNAs. (d) Fungus-aphid responsive target miRNAs. The horizontal axis represents the functional enrichment terms and the vertical axis represents the number of DEGs.

barley. This approach not only facilitates the identification of conserved regulatory elements despite evolutionary divergence but also enables us to infer the potential roles of barley genes based on their orthologs in Arabidopsis [57,58]. Although there are highly conserved molecular mechanisms between Arabidopsis and barley, due to their genetic distance, there will be differences in gene expression regulation that should be considered [56,59]. The presence of protected motifs was investigated using the MEME instrument with lengths ranging from 11 to 50 aa thus, a total of 11 motifs were selected in the DEG promoters.

We used the TOMTOM database to identify the predicted cis-regulatory factors linked to defense and hormone signaling. The separate analyses in the 4 meta-analyses suggested that BBR/BPC, tryptophan, AP2/EREBP, C2H2 zinc finger, bHLH, bZIP, and C4 zinc finger-type trans-acting factors play key roles in transcriptional activation and SAR/Induced Systemic Resistance (ISR) pathways. SAR and ISR are well-known plant defense mechanisms that

confer resistance against pathogens; SAR is typically activated by biotrophic pathogens and involves SA signaling, whereas ISR is triggered by beneficial microbes and relies on JA/ET signaling [17]. In addition, the results show that most of the CAREs have associated the C2H2 TFs, however, many elements including MA0120.1, MA0127.1, MA1073.1, MA0120.1, MA0255.1, and MA1255.1 remained unknown that might effect on aphid preference behavior for biotrophic/hemibiotrophic-infected hosts (S8 Table).

To study whether putative regulatory regions, spanning DEG promoters, are enriched with cis-acting elements, across-pathogens, an enrichment analysis of DEG motifs was conducted. For this purpose, the identified motifs were analyzed by GOMO. This tool analyzes a given motif to diagnose the GO categories linked to the DEGs of the binding motif (S9 Table).

The functional enrichment analysis of hormonal DEGs illustrated that these motifs play a role in regulating transcription (S10 Table). However, GO results on fungal DEGs demonstrated that these motifs play a role in regulating transcription, DNA-dependent, translation, and protein amino acid phosphorylation. Therefore, these motifs were involved in MFs, including TF function, serine/threonine kinase function, protein/ATP/nucleotide binding, ATP-dependent helicase activity, and structural components of the ribosome (S10 Table). Further, to study the possible biological roles of the aphid motifs, we identified the consensus mechanisms such as protein phosphorylation, regulating transcription, DNA-dependent, DNA replication, translation, phenylpropanoid synthesis, and transmembrane receptor tyrosine kinase signaling. The significant enriched elements were detected in MF, including TF function, protein serine/threonine kinase, microtubule motor function, and structural components of the ribosome (S10 Table).

Finally, GO enrichment analysis of fungus-aphid responsive cross-talk motifs was performed. These contain processes such as regulating transcription, DNA-dependent, translation, ovule development, leaf development, and transmembrane receptor tyrosine kinase signaling. The analysis of MF showed the motifs were involved in TF function, serine/threonine kinase function, protein/RNA/ATP binding, and structural components of the ribosome (S10 Table). These results, while enabling prediction of the variety of corresponding TFs, also facilitate their prioritizing and elucidation of their function in specific stages of transcriptional or defense response. As shown in Table S9 cis-acting elements are grouped into different functional categories, but interestingly this evaluation also highlighted the cis-element linked to JA (GO: 0009740), SA (GO: 0009751), and auxin pathways (GO: 0009733) in the 4 meta-analyses (S9 Table).

Since the cis-element is a sequence template that appears recurring in a series of DNA sequences, determining the most abundant sequences was considered in this study. Motifs 1, 6, and 2 were found across fungal, aphid-responsive, and hormonal signaling DEGs, but motifs 4 and 8 indicated the greatest prevalence in the fungus-aphid-responsive DEGs.

## Protein-protein interactions and selection of key hub

Since the primary objectives of this research are to investigate the relationship and impacts of fungal pathogens on pests and to study the hormonal signaling pathway in parallel, it is crucial to identify the DEGs that are regulated in this pathway. For this purpose, the network of co-expressed DEGs was developed to detect the key hub. Initially, to understand the degree of conservation in the protein-protein interactions in barley against pathogens and aphids, we presented a list of 89 common DEGs among four meta-analyses (Fig 9b).

Subsequently, an analysis of the protein-protein interaction (PPI) network was conducted utilizing the STRING web resource to reduce the number of interacting proteins and the complexity of the networks. In the end, only the network of the fungus-aphid-responsive genes and their hormonal partners was visualized using Cytoscape software (Fig 13). The giant network contained 88 nodes connected with 661 edges as illustrated. Some hub genes were viewed by using nodes having a large degree between centrality values, and or the genes

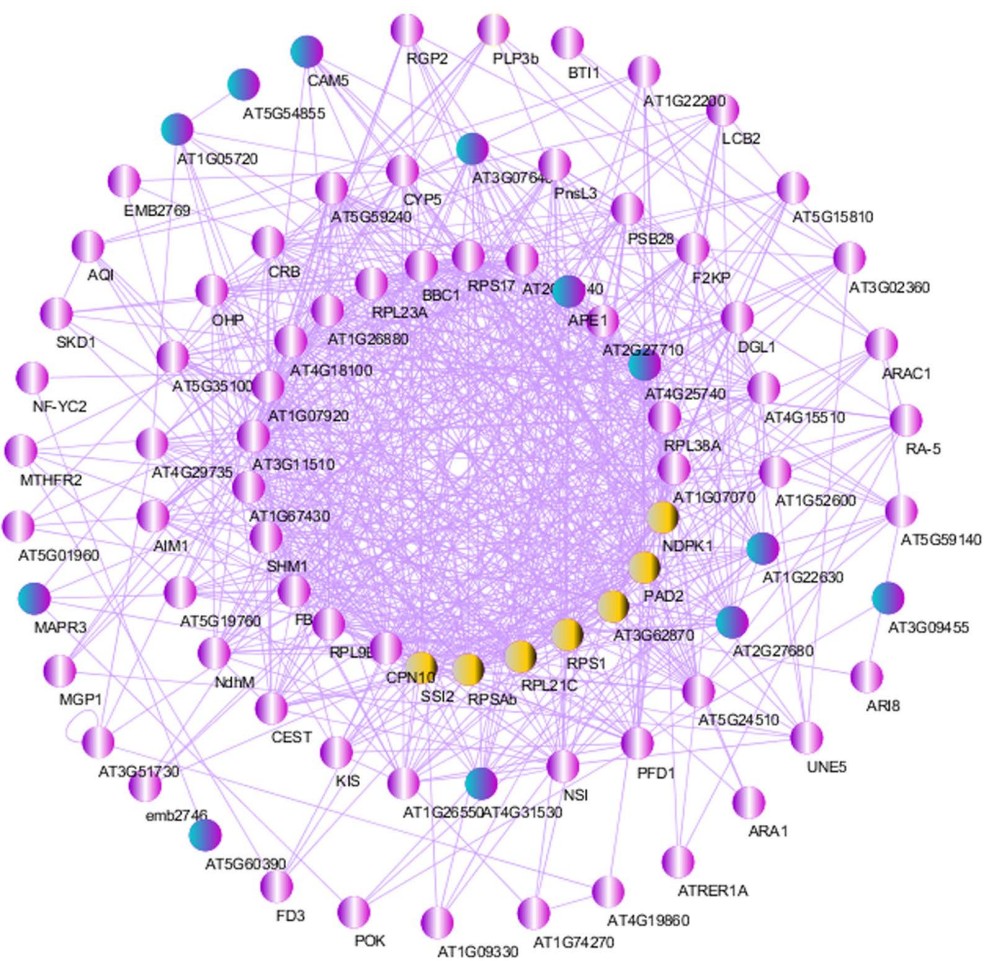

**Fig 13. The PPI network analysis through STRING web resource and Cytoscape.** This network illustrates the interactions between common hubs and DEGs of hormonal, fungal, aphid-responsive, and fungus-aphid-responsive. Internal nodes represent genes with more connections, while outer nodes indicate genes with fewer interactions. Yellow-highlighted nodes indicate a strong connection within the PPI.

with a high number of interactions (> 20). Including these genes can be mentioned *PAD2, SSI2, RPL21C, RPSAb, RPS1, NDPK1* and *AT3G62870* (yellow nodes). In addition, among the regulated proteins, it is worthy to notice some DEGs play a key role in the cross-talk between fungal-pests, and hormonal signaling in determining pathogenicity preference. Including these candidate genes can be mentioned *SSI2, PAD2, RPS1, RPS17, SHM1, CYP5, RPL21C, MAPR3, NF-YC2, AIM1*, and *FBA2*. We found 12 top candidate genes with unknown functions (turquoise nodes) including *AT4G31530, AT2G27680, AT1G22630, AT4G25740, AT1G05720*, and *AT5G54855* which might affect aphid preference behavior for biotrophic/hemibiotrophic-infected hosts (S11 Table). From these findings, we concluded that the DEGs demonstrate shared reactions to multiple stresses and could be valuable for characterizing overall resistance to stress in crops, especially barley.

## Leave-one-out cross-validation of meta-analysis

To assess the effectiveness of hub genes in pathogenicity, the Leave-one-out cross-validation (LOOCV) method was employed. The results (Fig 14) showed that identified

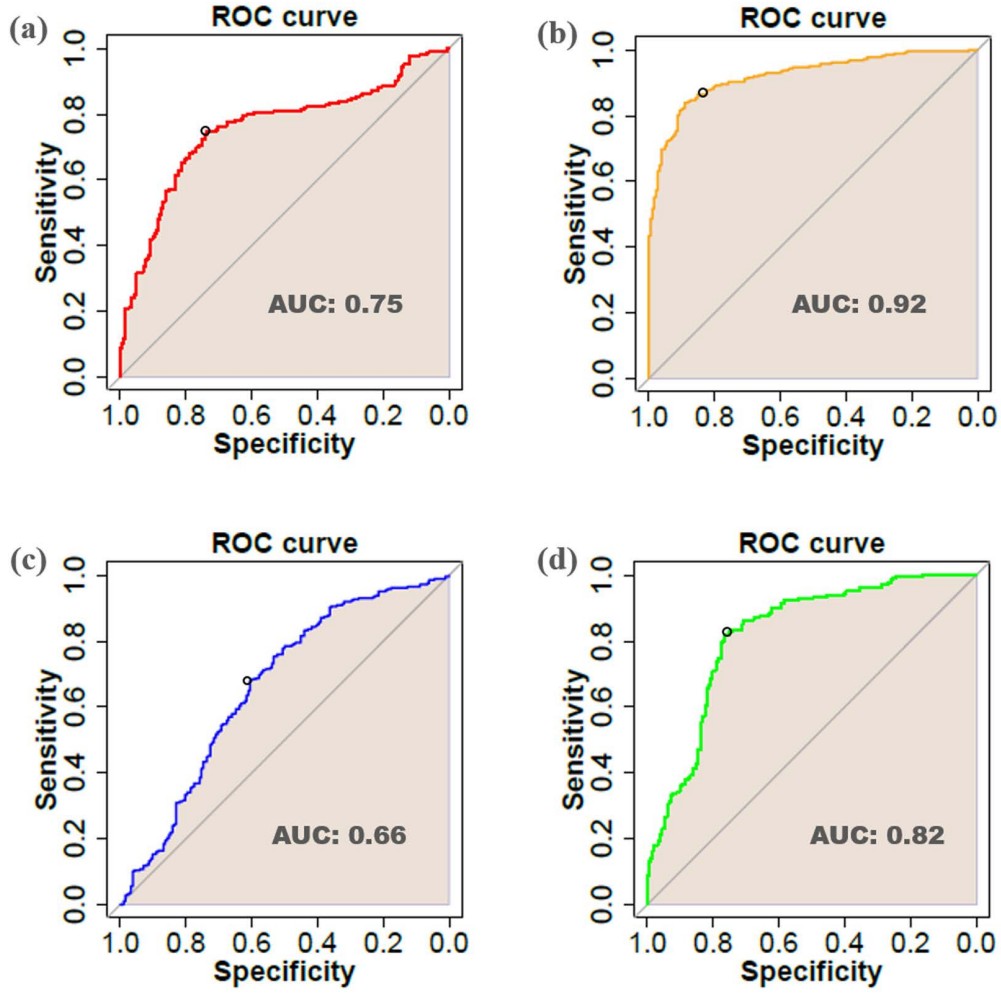

**Fig 14. Cross-validation analysis of ranked hub genes.** We estimate the classification error for each competitor using LOOCV. We present also the AUC area under the receiving operating characteristic (ROC) curve specific to each competitor in different colors. (a) ROC curve for the hormonal hub genes (red), (b) fungus hub genes (yellow), (c) aphid-related hub genes (blue), and d) fungus-aphid hub genes (green) are shown.

hub genes for hormonal, fungal, aphid, and fungus-aphid responsive datasets showed accuracy with AUC values of 0.75, 0.92, 0.66, and 0.82 respectively. This emphasizes the distinguishing capabilities of the detected hub genes and confirms the significance of the established DEGs (S3 Table).

## Discussion

Despite many studies, different aspects of the molecular mechanisms underlying host preference in barley for biotrophic/hemibiotrophic pathogens have remained unclear. For instance, what are the primary hormonal signaling pathways that mediate responses against fungal pathogens, and how do these pathways interact with insect herbivores? The integrative systems biology analysis of transcriptome provides valuable insights into plant-pathogen interactions, effectively addressing key biological questions.

## Gene ontology of DEGs, modules, and hub genes and identification of plant-pathogen interactions

During the ontology analysis of DEGs, modules, and hub genes, we identified several significant pathways related to host preference for biotrophic/hemibiotrophic pathogens as well as aphids (Figs 3 & 9, S2 Table), which provide insights into plant-pathogen interactions and their potential effects on aphid host preference. This analysis indicates that the plant's response to multiple stressors, including hormones and pathogens, might have direct or indirect effects on the host's preference for aphids. A wide range of BPs are regulated under pathogens attack, which may potentially play a role in the plant's response to both fungal pathogens and aphids and could affect the nutritional quality of host plants for aphids. For instance, the phytohormones SA and JA, as the hormonal backbone, constitute the plants' defense mechanisms against pathogenic attackers [60].

Thus plant responses to combined triggers are regulated by interacting metabolic and signaling pathways, which involve various physiological processes, such as photosynthetic activity, ROS scavenging, hormonal signaling, phenylpropanoid compounds, secondary metabolites, proteins, hypersensitive response (HR), toxic compounds accumulation, etc [61–64]. This response can affect the nutritional quality of host plants for aphids, potentially influencing their host preference. For instance, the production of secondary metabolites such as phenylpropanoid compounds is significant in plant fitness and has roles in growth and development, reproduction, and defense against stressors [65]. In interactions between host plants and invaders, phenolic compounds are thought to be elevated in response to damage caused by pests or pathogens. These phenolic compounds can have direct toxic effects on fungal pathogens by inhibiting their growth and reproduction.

Some studies have also shown that high levels of phenolics can make plants less attractive to aphids, which preferentially feed on plants with lower levels of phenolics. Overall, the role of phenolics in the host preference of aphids for plants infected with a fungal pathogen is complex and may be influenced by a variety of factors. Although high levels of phenolics may make plants less attractive to aphids, they also contribute to plant defense against fungal pathogens [66]. Similarly, the accumulation of toxic compounds can affect the fitness of both the pathogens and aphids. Alcohols, aldehydes, and esters are organic compounds that are constantly released from nearly all plants upon mechanical damage and pathogen attacks [67]. They can activate defense mechanisms to improve defense against specific foliar- and phloem-feeding insects and necrotrophic fungi [68].

As a result, the releasing these compounds, plants can more effectively stressors, which leads to a decreased host preference for pathogens and aphids [68]. The role of phytohormones in plant defenses against pathogens and aphid host preference has been extensively studied. These hormones mediate plant defense mechanisms and offer insights into their underlying molecular processes. Aphid feeding on host plants induces several plant defense signaling pathways, and many of these signaling cascades respond to changes in hormonal levels of ABA, SA, JA, and aphid feeding-induced cytokinins, which contribute to enhanced photosynthesis, growth, and tolerance. For example, SA is a key defense hormone that mediates immunity against biotrophic and hemibiotrophic pathogens that are effective against herbivores, such as aphids, as they feed better on plants with reduced JA defenses. Conversely, plants essentially induce the JA pathway in resistance to insect attack and infection by necrotrophic pathogens [69]. However, once the hemibiotrophic pathogen infects tissue and then switches to necrotrophic feeding, JA-mediated defenses are activated. This can make the plant inhospitable to aphids, which prefer plants with lower JA-mediated defenses [70]. The interactions between JA/SA signaling are believed to equip plants with adaptive

mechanisms that optimize resource allocation while influencing host preference dynamics for aphids [18].

## TFs involving in cross-talks of fungus-aphid and host as potential plant repressors/activators

Barley TFs related to host preferences for fungal and pest pathogens are also presented. Consequently, TFs act as repressors or activators, either alone or with other proteins, forming multimeric complexes [71]. Among the identified TF families, MADS-box, AP2/ERF, and C2H2-ZF play key roles in plant resistance to biotrophic/hemibiotrophic pathogens, as well as in host preference for aphids (Fig 10a, S4 Table). The MADS-box family is identified based on a conserved motif of 60-amino-acid within the DNA-binding domain at the N-terminal of the sequence [72]. Among genes that are specifically activated or repressed against aphids or fungi, MADS-boxes are probably the most studied family of transcriptional regulators because of their significant roles in pathogen defense[73]. In hosts infected with biotrophic pathogens that suppress JA-mediated defenses, MADS-box TFs can be induced by SA signaling to regulate defense responses [74].

However, this defense mechanism may inadvertently make the plant more suitable for aphid feeding. On the other hand, in plants infected with hemibiotrophic pathogens, MADS-box TFs can be induced by JA signaling to regulate defense responses [75], which may concurrently reduce plant host preference for aphids. Therefore, the role of MADS-box TFs in the host preference of aphids for plants infected with biotrophic and hemibiotrophic pathogens is complex. The signaling pathways of MADS-box TFs can either enhance or reduce a plant's suitability as a host for aphids, depending on the pathogen and the plant's defense response.

The second TF family associated with host preference for pathogens and pests is AP2/ERF. This family was one of the largest classes of potential TF and plays a crucial role in various regulatory processes, including complicated developmental processes, leaf senescence, fruit ripening, resistance to pathogen attack, and responses to plant hormones such as SA and JA/ET [76]. Studies have shown that some members of the AP2/ERF TF family act as positive or negative regulators of plant immunity and can be manipulated by biotrophic/hemibiotrophic pathogens to promote their growth. This manipulation can increase the host plant's susceptibility to aphid feeding. Thus, TFs that regulate the JA/ET signaling pathway, such as *ERF1* and *ERF2*, have been shown to activate defense responses against biotrophic/hemibiotrophic pathogens, thus reducing plant susceptibility to aphid feeding. In some cases, AP2/ERFs regulate disease resistance by simultaneously modulating SA and JA biosynthesis. For example, *ERF3* positively regulates resistance to herbivores by increasing the accumulation of SA, JA, and ET [77].

The third TF family associated with host preference for pathogens and pests is the zinc finger (ZF) family. C2H2 zinc finger proteins are among the most abundant TFs investigated in plants and exhibit diverse cellular functions, including transcriptional regulation, oxidative stress signals, plant development, stress response, hormonal signal transmission, and PPI [78,79]. The analysis of DEG ontology and promoter motifs revealed that the majority of *C2H2-ZF* genes participate in various molecular functions, such as metal ion binding and responses to different stimuli (S10 Table). The C2H2 (Zn) family plays a central role as a key transcriptional suppressor involved in plant defense responses, especially in biotrophic and necrotrophic pathogens [80]. Specifically, some of their downstream targets are involved in the SA signaling pathway, which is important for the defense against biotrophic pathogens. In contrast, during infection with hemibiotrophic pathogens, some members of the C2H2-ZF TF family can regulate JA and ET signaling pathways, which are critical for defense against

necrotrophic fungi, resulting in reduced host preference for aphids [79,81]. However, the role of these proteins in host preference for aphids remains complex and requires further investigation [82]. Our results suggest that these TFs can regulate SA, JA, and ET signaling pathways that are critical for defense against biotrophic/hemibiotrophic pathogens, consequently reducing host preference for aphids.

## Protein kinases and their crucial function in the regulation of immunity

The 22 families of kinases by the iTAK tool were recognized in barley (Fig 10c, S6 Table). Protein kinases are key messengers in signaling networks that regulate and coordinate different cellular functions, such as transcription, metabolism, cell cycle, and communication. In addition, most PKs play a critical role in the regulation of immune response [83]. Among them, a conserved family of protein kinases belonging to the tyrosine kinase-like kinase (TKL) group has been linked to the signaling pathways implicated in defense against pathogens, growth, development, and production of pro-inflammatory factors [84]. Common RLK/Pelle comprises the ligand-receptor complex and PK domains that play a role in downstream signaling. The number of RLK/Pelle are introduced to act as designated microbe-associated molecular patterns (MAMPs) receptors. Significant examples include the Xa21 tolerance DEGs of rice to *Xanthomonas campestris* pv. oryza) and pathogen-associated molecular pattern (PAMP)-receptor FLS2 in *A. thaliana* identify bacterial flagellin [85,86].

Overall, RLK/Pelle-type genes are likely essential for primary pathogen detection as PAMP/MAMP receptors and for the induction of downstream immune responses, such as the production of ROS and the regulation of tolerant-related genes [85]. Some studies suggest that RLKs, including Pelle, may contribute to the regulation of plant immunity against biotrophic pathogens, which could indirectly affect aphid host preference [87]. These responses can restrict the growth and development of biotrophic pathogens, which in turn can affect the quality of the host for aphids [87]. In addition to RLK-Pelle, the most numerous kinase groups were CAMK, TKL, and AGC (Fig 10c). Overall, 95% of barley kinomes consist of these PKs. The other groups, which accounted for 5% of the entire barley kinome, each contained a limited number of loci (≤16). CAMKs are known for their role in calcium signaling pathways that regulate plant immune responses. Some studies indicate that CAMKs, such as *CYP73A5*, are involved in the regulation of plant defense responses against biotrophic pathogens, which could indirectly affect aphid host preference [88–90]. *CYP* genes play a critical role in the production of phenolic acids, highlighting their conserved function in the biosynthesis of phytohormones and secondary metabolites and mediating plant responses to both biotic and abiotic stresses [53,54].

However, the specific mechanisms by which CAMKs may affect aphid host preferences are not clear. AGC protein kinases, such as *OPR2*, play a significant role in the regulation of JA and SA functions, which are important in the plant defense response against both biotrophic and hemibiotrophic pathogens. Overall, the specific roles of CAMKs, TKLs, RLK/Pelle, and AGC protein kinases in the host preference of aphids in plants infected with biotrophic and hemibiotrophic pathogens are likely to be complex and multifaceted. Further research is required to fully understand the involvement of these interactions.

## Micrornas (Mirnas) as regulators of ETI and PTI

In various plant species, miRNA families play potential roles in regulating defense genes associated with plant immunity, particularly in biotrophic and hemibiotrophic pathogens [91]. The miRNA targets play a role in regulating PTI and ETI responses. PTI is a primary immune response triggered by the recognition of common PAMPs by PRRs on the cellular surface and

involves hormone signaling, ROS evolution, callose deposition, and other mechanisms [92]. ETI is a stronger and more specific immune response that specially regulates resistance (R) gene expression [92]. miRNAs appear to regulate these R-gene-produced transcript targets, however, they cause simple transcript cleavage in many instances. Furthermore, recent studies have indicated that miRNAs not only regulate local defense responses but may also influence SAR [17], thereby affecting the overall fitness and susceptibility of host plants to aphids and pathogens. Understanding the intricate roles of miRNAs in these processes is essential for elucidating host-pathogen interactions and improving disease resistance in crops.

According to the results, many known miRNA groups, including hvu-miR156, hvu-miR159, hvu-miR168, hvu-miR6198, hvu-miR397, hvu-miR171, hvu-miR6195, hvu-miR5049, hvu-miR6199, and hvu-miR6181 were identified as being associated with barley responses to pathogens and herbivores (Fig 11, S7 Table). These miRNAs were found to vary mostly before and after the disease barley leaf stripe (BLS), suggesting that they are associated with immunity to biotrophic/hemibiotrophic pathogens in barley [93]. The identified miRNAs are believed to play crucial roles in regulating hormonal signaling pathways involved in plant defense responses to biotic and abiotic stimuli. For example, *miR5049* target genes were linked to the ABA ripening protein, which is a critical regulator in response to various stressors [94,95] and could also affect host preference for pathogens. *miR5049* targets transcripts from the oxidoreductase family, specifically short-chain dehydrogenase/reductase (SDRs), as well as UDP-glucoronosyl and UDP-glucosyl transferase domain transcripts, which are significant in various defense processes [96]. *miR6192* targets a gene involved in the mitogen-activated protein kinase (MAPK) process, which is crucial for transmitting signals from cell receptors to DNA in the cell nucleus and immune-related mechanisms [97].

In barley, some miRNAs are differentially expressed in response to biotrophic and hemibiotrophic pathogens. For instance, *hvu-miR156* and *hvu-miR159* are down-regulated in resistance to the biotrophic fungus *Blumeriagraminis f. sp. hordei*, while *hvu-miR168* and *hvu-miR6198* were up-regulated [98]. Similarly, *hvu-miR397* was up-regulated in resistance to the hemibiotrophic fungus *Fusarium culmorum* [99]. The differential expression of these miRNAs might affect the quality of host plants for aphids by regulating defense responses against these pathogens [100]. For instance, the down-regulation of *hvu-miR156* and *hvu-miR159* in response to *B. graminis* infection could lead to the up-regulation of their target genes, which are involved in the regulation of plant survival and development [98]. The up-regulation of *hvu-miR397* in response to *F. culmorum* infection could lead to the down-regulation of its target DEGs, which play a role in the production of lignin and other defense-related compounds [99,101]. This could potentially affect the resistance of the host plants to aphids. Furthermore, miRNAs have been demonstrated to contribute to the regulation of plant immune-related genes, including those that participate in the generation of secondary metabolites and the induction of defense functions [98,102]. These responses can limit the growth and development of biotrophic and hemibiotrophic fungi, which may indirectly affect the quality of host plants for aphids [98]. Further research is needed to understand the specific roles of barley miRNAs in host plant-aphid interactions.

## Selective hub genes and host preference

Three hub genes, namely *PAD2*, *RPS1*, and *SSI2,* were identified (Fig 13) as key candidates for further exploration regarding their roles in host-pathogen interactions. Consequently, we will focus on these significant genes to elucidate the specific molecular pathways that influence plant host preference in response to biotrophic/hemibiotrophic pathogens as well as herbivores.

## Phytoalexin-deficient 2 (*PAD2*)

Plants often respond to pathogen or insect attacks by inducing the synthesis of toxic compounds such as phytoalexins and glucosinolates (GS). *PAD2* encodes γ-glutamylcysteine synthase, which is involved in glutathione biosynthesis (GSH), a key antioxidant in plants. GSH is essential for maintaining cellular redox homeostasis and is involved in the production of secondary metabolites that enhance disease resistance. Studies have shown that the *pad2* mutant exhibits significantly reduced levels of GSH — approximately 20% of wild-type levels—leading to increased susceptibility to various pathogens, including biotrophic fungi [103]. Moreover, *PAD2* has been implicated in signaling pathways involving SA, which is a key regulator of plant defense responses. Notably, the accumulation of SA and pathogenesis-related proteins is diminished in *pad2* mutants, indicating that *PAD2* is not merely a biosynthetic component, but also a regulatory factor in defense signaling [104].

Furthermore, GSH deficiency in *pad2* plants correlates with reduced levels of GS, which is important for insect resistance. Specifically, lower GSH levels in *pad2* mutants correlated with reduced accumulation of indole and aliphatic GSs upon insect feeding, leading to increased susceptibility to herbivorous insects [105]. This suggests that *PAD2* not only affects pathogen resistance but also plays an essential role in mediating plant responses to herbivory through its effects on GSH and GS biosynthesis. Thus, *PAD2* has emerged as a key player in linking hormonal signaling and metabolic pathways that govern host preference against both pathogens and herbivores [104,105].

## Ribosomal protein S1 (*RPS1*)

Ribosomal proteins (*RPS*) constitute the protein component of ribosomes and play a significant role in ribosome biogenesis, protein synthesis, cell growth, and apoptosis [106]. Isolating *RPS1* (30S ribosomal protein S1) has been shown to prevent biotrophic fungal pathogens such as blight-infected leaves [107]. A group of *RPS*, such as *RPL*, has been identified as an important gene responsible for insect resistance in rice [108]. Thus, *RPS* activity enhances resistance, resulting in decreased host preference for biotrophic pathogens and aphids [107,108]. *RPS1* may form complexes with other R proteins, such as *RPS4*, which interacts with the WRKY domain of *RPS1*, triggering RRS1-mediated ETI. Additionally, *RPS4* interacts with TFs, such as *WRKY33*, *WRKY41*, *WRKY60*, and *WRKY70*, disrupting WRKY-mediated immune responses [109]. The expression of several *RPS* genes, especially *RPS1, RPS17, RPS13a,* and *RPL30,* is also negatively regulated by phytohormones, ABA, and cytokinins [106]. Further exploration of these DEGs, along with lateral R genes from other crop species, will enhance our understanding of the evolution and function of R genes [110].

## Salicylic acid Insensitive2 (*SSI2*)

Arabidopsis *ssi2* is a mutant line that restores SA signaling in the *npr1* genetic background. *SSI2* encodes a plastid-localized stearoyl-acyl carrier protein desaturase (SACPD) that desaturates stearic acid to oleic acid in chloroplasts [111]. The *ssi2* mutant exhibits several phenotypic traits, including dwarfing, accumulation of high levels of SA, constitutive overexpression of *PR1*, enhanced resistance to pathogens, and spontaneous cell death [17]. While the *ssi2* mutation leads to elevated levels of SA — promoting resistance against biotrophic pathogens and herbivorous insects — it concurrently impairs JA signaling pathways. This impairment results in increased susceptibility to necrotrophic pathogens, such as *Botrytis cinerea*, as JA is essential for activating defense responses against such threats [112].

Although *SSI2* catalyzes the desaturation step required for JA biosynthesis, the ssi2 mutation does not alter the levels of the JA precursor oleic acid and cannot be rescued by

exogenous JA. This shift is critical because while SA-mediated defenses are typically effective against biotrophs, necrotrophic pathogens require JA-derived signals for effective resistance. Therefore, increased levels of SA in the *ssi2* mutant may influence host preference by making the plant less attractive to biotrophic pathogens and herbivores [17,113]. The intricate relationship between these two hormone pathways underscores a complex regulatory network in which SA can suppress JA signaling, thereby influencing overall plant immunity and host preference dynamics [17].

## Conclusions

There are remarkably few studies investigating how plant pathogenic fungus and hormonal signaling may interact and impact the preference and performance of the *Rhopalosiphum padi*. We elucidated the complex interactions between barley plants and biotrophic/hemibiotrophic pathogens, highlighting critical biological and hormonal pathways that influence host preference and performance. Notably, 70% of the common DEGs were uniquely regulated by JA/SA signaling pathways, while 30% were co-regulated by both hormones, suggesting a nuanced interplay in plant defense mechanisms. We identified significant transcription factor-binding sites, particularly for AP2/ERF and C2H2 zinc finger factors, which are pivotal in the SAR and ISR pathways. Furthermore, we identified hub genes with unknown functions, such as *AT4G31530*, *AT5G54855*, *AT3G07640*, and *AT1G05720,* as promising candidates for future research in the context of influencing plant defense strategies and preferences against herbivory and infection. This research effectively integrated available specified microarray data, but the lack of RNA-seq data under similar conditions forced us to accept the available data. This underscores the necessity for further exploration and integration of diverse data sources to fully understand host-pathogen interactions and hormonal responses in barley. A key area for future research will be to assess the comparative significance of fungi on tripartite cross-talk, in comparison to other abiotic and biotic factors that impact plant-aphid cross-talk and hormonal signaling.

## Supporting information

**S1 Fig. Reducing heterogeneity among samples of studies for direct merging meta-analysis** . (a) The plot box of the E-GEOD-20279 dataset related to pathogens stresses in the Affymetrix platform with 3 control samples and 3 treatment samples, was drawn in the pre-normalization stage. (b) The plot box of the E-GEOD-20279 dataset after normalization, where all comparisons that are not significant or are not equal to the change threshold are converted to a log 2 value to remove a possible error. This method ensured that weak expression fluctuations were more likely to be real biological signals than measurement errors or errors not corrected by RMA normalization. The biological errors and batch effects have been corrected. After preprocessing, the black lines of the box plot are almost on the same straight line, indicating a high level of normalization. The horizontal axis stands for the control and treatment of different samples, while the vertical axis represents the expression value. The black line in the box represents the expression median for each sample.
(DOCX)

**S1 Table. List of identified DEGs by meta-analysis of fungal, pests, fungal-pests stress, and hormonal data and their corresponding UniGene, UniProt, and TAIR IDs.**
(XLSX)

**S2 Table. Gene ontology annotation of modules found by the Kegg pathway of fungal, pests, fungal-pests stress, and hormonal data.**
(XLSX)

**S3 Table. List of top 30 hub genes in each module for fungal, pests, fungal-pests stress, and hormonal data.**
(XLSX)

**S4 Table. List of transcription factors enriched for fungal, pests, fungal-pests stress, and hormonal DEGs.**
(XLSX)

**S5 Table. List of transcription regulators enriched for fungal, pests, fungal-pests stress, and hormonal DEGs.**
(XLSX)

**S6 Table. List of protein kinase enriched for fungal, pests, fungal-pests stress, and hormonal DEGs.**
(XLSX)

**S7 Table. List of the highest scoring miRNAs associated with fungal, pests, fungal-pests stress, and hormonal DEGs that were identified using psRNA target server.**
(XLSX)

**S8 Table. List of known cis-acting elements fungal, pests, fungal-pests stress and hormonal DEGs by Tomtom search against the JASPAR database.**
(XLSX)

**S9 Table. Significant GO term enriched by GOMO analysis for fungal, pests, fungal-pests stress, and hormonal DEGs.**
(XLSX)

**S10 Table. The conserved cis-acting elements found in promoters of fungal, pests, fungal-pests stress, and hormonal DEGs by the MEME analysis.**
(DOCX)

**S11 Table. Identification of hub genes in all 4 datasets with unknown biological functions using the UniProt database.**
(XLSX)

# Author contributions

**Conceptualization:** Ali Moghadam.

**Data curation:** Zahra Soltani, Ali Moghadam, Mohammadreza Shamekh.

**Formal analysis:** Zahra Soltani, Mohammadreza Shamekh.

**Investigation:** Zahra Soltani.

**Methodology:** Zahra Soltani, Mohammadreza Shamekh.

**Project administration:** Ali Moghadam.

**Software:** Zahra Soltani.

**Supervision:** Ali Moghadam.

**Validation:** Ali Moghadam.

**Visualization:** Mohammadreza Shamekh.

**Writing – original draft:** Mohammadreza Shamekh.

**Writing – review & editing:** Ali Moghadam.

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
