## [Decision Letter · Decision Letter 0]

30 Oct 2024

PONE-D-24-35105Comparative meta-analysis of barely transcriptome: pathogen type determines host preferencePLOS ONE

Dear Dr. Moghadam,

Thank you for submitting your manuscript to PLOS ONE. After careful consideration, we feel that it has merit but does not fully meet PLOS ONE’s publication criteria as it currently stands. Therefore, we invite you to submit a revised version of the manuscript that addresses the points raised during the review process.

We look forward to receiving your revised manuscript.

Kind regards,

Sumit Jangra, Ph.D.

Academic Editor

PLOS ONE

**Journal Requirements:**

Reviewers' comments:

Reviewer's Responses to Questions

**Comments to the Author**

1. Is the manuscript technically sound, and do the data support the conclusions?

Reviewer #1: Partly

Reviewer #2: Partly

2. Has the statistical analysis been performed appropriately and rigorously? 

Reviewer #1: Yes

Reviewer #2: Yes

3. Have the authors made all data underlying the findings in their manuscript fully available?

Reviewer #1: Yes

Reviewer #2: Yes

4. Is the manuscript presented in an intelligible fashion and written in standard English?

Reviewer #1: No

Reviewer #2: No

5. Review Comments to the Author

**Reviewer #1: ** Summary

The authors present a meta-analysis and co-expression network analysis of barley transcriptomic responses to fungal infections, aphid infestations, and hormone treatments, using microarray data. The study highlights interesting hub genes, transcription factors, cis-regulatory elements, and miRNAs, along with associated GO terms. The authors suggest that their findings offer insights into barley's crosstalk between biotic stresses and introduce candidate genes for potential genetic engineering applications.

Major Points

1. The manuscript was provided without line numbers, making it difficult to refer to specific sections during the review. Please ensure line numbers are added in future submissions.

2. While the current analysis is valid for the datasets used, it would be ideal to incorporate RNA-seq data, which would provide higher resolution and greater dynamic range compared to microarrays. This could either be incorporated into the meta-analysis and subsequent analyses or could be used as validation for the conclusions made. If relevant RNA-seq data is not available for barley under similar conditions, this limitation should be acknowledged in the discussion.

3. Given that the study is based on a relatively small gene set (“approximately 2,000 genes”), the scope of biological conclusions drawn from the GO analysis may be limited. I suggest the authors clarify the limitations of their findings and consider expanding the study with additional data, if possible, or tempering the claims made based on this limited set.

4. Barley and Arabidopsis are evolutionarily distant, and while Arabidopsis serves as a useful model, there are significant differences in gene regulation and promoter architecture between the species. The use of Arabidopsis promoters to infer functionality in barley is problematic, as promoter elements can differ even within related plant clades. I recommend the authors acknowledge this as a significant limitation and be cautious when making strong functional inferences based on Arabidopsis orthologs and promoter sequences.

5. The discussion section is overly long and frequently references other studies that are not directly related to the data presented in this manuscript. For instance, the section on GLVs (green leaf volatiles) introduces concepts that are not mentioned earlier in the paper, making it seem disconnected from the main results. Furthermore, claims such as "This meta-analysis confirmed that fungi induce more GLVs than insects" and "our overall meta-analysis shows that crops…result in notably elevated levels of phenolic compounds…" are not supported by the data, as the study does not quantify these compounds but only examines gene expression related to their production. Please revise the discussion to focus more on your study's findings.

6. I appreciate that the authors are non-native English speakers; however, certain sections, particularly the abstract and introduction, are difficult to read due to incorrect sentence structure, grammar, and punctuation. I recommend that the manuscript be reviewed by a native English speaker or a professional editor to improve clarity and readability.

Minor Points

- There are numerous typographical errors throughout the manuscript. However, without line numbers, it is difficult to provide specific examples. Please proofread carefully.

- Figure 1 is overly complex and lacks clarity. For example, no explanation is provided to differentiate the black boxes from the purple ellipses. According to the methodology, DEGs from the meta-analysis and co-expression analysis should feed into subsequent analyses (lateral analysis, GO enrichment), but this is not shown in the figure. Moreover, it seems to suggest that only "Fungi" data is used for co-expression analysis, only "Insect" for network analysis, and both "Insect & Fungi" for cis-acting element analysis, which contradicts the text. I recommend reformatting this figure to make the workflow clearer and more legible.

- Figure 2 is not directly referenced in the text, causing the subsequent figure numbering to be out of sync. Please ensure all figures are referenced in the text and check that the correct figure numbers are used. Also, I find it unclear how upregulated and downregulated DEGs can overlap in this figure.

- The number of genes associated with each GO term in Figure 4 does not match the values mentioned in the text. For instance, "protein folding" shows fewer than 25 genes in the figure 4b, though 7% of 2,531 genes should yield around 177 genes. Similarly, "response to reactive oxygen species" shows 50+ genes in the figure 4c, but according to the text, 2% of 62 DEGs is approximately 1 gene. These inconsistencies need to be corrected for accuracy.

- The "PRISMA 2020 checklist" supplementary file contains important information about dataset adherence to MIAME guidelines, which is not addressed in the main manuscript. This information should be incorporated into the manuscript, as it is relevant to the quality of the data used.

**Reviewer #2:**  I think that the choice of research content is more meaningful. However, there are a lot of errors in the content, and it seems that the authors submitted this manuscript without checking it carefully, for example, there is an intersection of the Venn diagrams of the up- and down-regulated genes, and Fig2 in the manuscript actually refers to Fig3. In addition, the discussion section, although it is written a lot, it is not focused, and does not discuss the results of their own research, it is more like a review.

6. PLOS authors have the option to publish the peer review history of their article (what does this mean? ). If published, this will include your full peer review and any attached files.

**Do you want your identity to be public for this peer review?** For information about this choice, including consent withdrawal, please see our Privacy Policy .

Reviewer #1: No

Reviewer #2: No

---

## [Author Response · Author response to Decision Letter 1]

26 Dec 2024

Dear Editor in chief

Thank you in advance for your considerations. We answered all of reviewers’ comments here.

Best regards,

Reviewer #1: Summary

The authors present a meta-analysis and co-expression network analysis of barley transcriptomic responses to fungal infections, aphid infestations, and hormone treatments, using microarray data. The study highlights interesting hub genes, transcription factors, cis-regulatory elements, and miRNAs, along with associated GO terms. The authors suggest that their findings offer insights into barley's crosstalk between biotic stresses and introduce candidate genes for potential genetic engineering applications.

Major Points

1. The manuscript was provided without line numbers, making it difficult to refer to specific sections during the review. Please ensure line numbers are added in future submissions.

Answer: Line numbers were added in the revised version of the manuscript.

2. While the current analysis is valid for the datasets used, it would be ideal to incorporate RNA-seq data, which would provide higher resolution and greater dynamic range compared to microarrays. This could either be incorporated into the meta-analysis and subsequent analyses or could be used as validation for the conclusions made. If relevant RNA-seq data is not available for barley under similar conditions, this limitation should be acknowledged in the discussion.

Answer: We really appreciate you for your comment. Actually, we recognized some RNA-seq data, but they are not similar to our microarray conditions that we explained this issue based on your comment in the conclusion. We mentioned some detailed reasons here:

- We have four distinct microarray datasets (fungal, hormonal, pest, and fungal-pest interactions) and the lack of similar RNA-seq data makes the comparison impossible.

- We have analyzed rather big data and despite their limitations, the results remained statistically significant in different analyses. They have shown specific transcriptomic responses of barley to the target pathogens.

- In addition, the results demonstrate meaningful statistical relevance, supported by robust cross-validation methods detailed in the manuscript. This validation (LOOCV) approach ensures that our findings are reliable.

Despite our extensive validation analyses, we acknowledge that the lack of relevant RNA-seq data under similar conditions is a limitation. We have addressed these in the conclusion section to provide a comprehensive understanding of the limitations faced in our research.

3. Given that the study is based on a relatively small gene set (“approximately 2,000 genes”), the scope of biological conclusions drawn from the GO analysis may be limited. I suggest the authors clarify the limitations of their findings and consider expanding the study with additional data, if possible, or tempering the claims made based on this limited set.

Answer: Thank you for your valuable comment. In the manuscript, we have mistakenly stated, "Each dataset contains contained more than 16,300 probeset IDs representing approximately 2,000 genes." The correct sentence is: "Each dataset contains more than 20,000 probeset IDs representing approximately 7,000 genes." We corrected this accidentally mistake in the text. After normalization of the datasets, we did a series of analyses and filtering based on the P-value or FDR ≤ 0.05 mentioned in the text, and we identified a set of differentially expressed genes (DEGs) that are statistically significant. Finally, the GO analysis has shown statistically significant pathways specifically related to the target pathogens and molecular responses. We acknowledge that using a smaller gene set may limit some biological conclusions drawn from GO analysis. In addition, we have addressed the possible limitations in the manuscript based on your comment and emphasized that is better to do further research with additional datasets to enhance the result robustness.

4. Barley and Arabidopsis are evolutionarily distant, and while Arabidopsis serves as a useful model, there are significant differences in gene regulation and promoter architecture between the species. The use of Arabidopsis promoters to infer functionality in barley is problematic, as promoter elements can differ even within related plant clades. I recommend the authors acknowledge this as a significant limitation and be cautious when making strong functional inferences based on Arabidopsis orthologs and promoter sequences.

Answer: Thank you for your valuable feedback.

Actually, we have a big list of candidate genes used to promoter analysis. This impossible to analysis those individually. Therefore, we have to analysis these genes as a list. In addition, the practical databases like BioMart and GOMO show the best results based on the Arabidopsis and if we use the barley IDs as impute, we lost a lot of data. Of course, we have chosen the highly conserve regulatory motifs that are significantly specific relationships with target pathogens and molecular responses in barley. Although many authoritative papers have used the same approch, we have outlined possible limitations.

Anyway, we improved the text based on your comment and acknowledged the possible limitations in the text. We have addressed that functional roles of regulatory cis-acting elements derived from Arabidopsis may not accurately predict their roles in barley due to these evolutionary distinctions. Therefore, we recommend a cautious interpretation of results derived from these analyses in our manuscript. We hope this additional context will satisfy the reviewer's concerns. We aim to bridge the gap in knowledge and provide a foundation for future research that can validate these findings in the context of barley-specific biology.

In addition, some detailed reasons are here:

- Arabidopsis has a well-annotated genome with a vast array of functional data, which can provide insights into gene functions and regulatory mechanisms that may not yet be available for barley. This enables us to infer potential roles of barley genes based on their orthologs in Arabidopsis.

- Despite the evolutionary differences, certain promoter motifs and transcription factor binding sites are conserved across species. Research has demonstrated that many transcriptional regulatory mechanisms remain analogous, enabling meaningful insights into barley gene expression by examining their Arabidopsis counterparts.

5. The discussion section is overly long and frequently references other studies that are not directly related to the data presented in this manuscript. For instance, the section on GLVs (green leaf volatiles) introduces concepts that are not mentioned earlier in the paper, making it seem disconnected from the main results. Furthermore, claims such as "This meta-analysis confirmed that fungi induce more GLVs than insects" and "our overall meta-analysis shows that crops…result in notably elevated levels of phenolic compounds…" are not supported by the data, as the study does not quantify these compounds but only examines gene expression related to their production. Please revise the discussion to focus more on your study's findings.

Answer: Thank you, we removed unrelated explanations that did not align with the main objectives of the manuscript. Additionally, we have condensed and specified the discussion to focus more directly on our results.

6. I appreciate that the authors are non-native English speakers; however, certain sections, particularly the abstract and introduction, are difficult to read due to incorrect sentence structure, grammar, and punctuation. I recommend that the manuscript be reviewed by a native English speaker or a professional editor to improve clarity and readability.

Answer: Thank you, we appreciate your understanding that we are non-native English speakers. In response to your comments, we have thoroughly revised the manuscript, focusing specifically on the abstract and introduction. We have made significant improvements to the sentence structure, grammar, and punctuation to enhance clarity and readability.

Minor Points

- There are numerous typographical errors throughout the manuscript. However, without line numbers, it is difficult to provide specific examples. Please proofread carefully.

Answer: Thank you for your feedback. We have carefully proofread the manuscript and corrected all typographical and writing errors. Additionally, we have included line numbers in the revised version to facilitate easier reference for any future comments or suggestions.

- Figure 1 is overly complex and lacks clarity. For example, no explanation is provided to differentiate the black boxes from the purple ellipses. According to the methodology, DEGs from the meta-analysis and co-expression analysis should feed into subsequent analyses (lateral analysis, GO enrichment), but this is not shown in the figure. Moreover, it seems to suggest that only "Fungi" data is used for co-expression analysis, only "Insect" for network analysis, and both "Insect & Fungi" for cis-acting element analysis, which contradicts the text. I recommend reformatting this figure to make the workflow clearer and more legible.

Answer: Thank you for your exact comments. We replaced the figure with a simple and comprehensive flowchart to clear up the confusion. Additionally, I have ensured that the workflow accurately reflects the methodology, illustrating how the DEGs identified from meta-analysis and the modules derived from co-expression analysis feed into GO enrichment. PPI interaction network, and cross-validation were performed for the co-expression modules identified. We have also clearly indicated that DEGs obtained from the meta-analysis are utilized for lateral analyses, such as identifying transcription factors, protein kinases, microRNAs, and ultimately for cis-acting element analysis.

- Figure 2 is not directly referenced in the text, causing the subsequent figure numbering to be out of sync. Please ensure all figures are referenced in the text and check that the correct figure numbers are used. Also, I find it unclear how upregulated and downregulated DEGs can overlap in this figure.

Answer: Thank you for your exact comment. Figure 2 is one step of data processing and checking during the meta-analysis and before the filtering of DEGs, thus does not represent the final results. This figure is accidentally presented and no need to present as a main figure. In this Venn diagram we checked the possible overlap between the up-regulated and down-regulated DEGs and have selected each overlap up or down gene based on the lowest and most significant P-value, in this case there finally will be no overlap. We appreciate your understanding and will ensure that all figures are correctly referenced and numbered in the revised submission.

- The number of genes associated with each GO term in Figure 4 does not match the values mentioned in the text. For instance, "protein folding" shows fewer than 25 genes in the figure 4b, though 7% of 2,531 genes should yield around 177 genes. Similarly, "response to reactive oxygen species" shows 50+ genes in the figure 4c, but according to the text, 2% of 62 DEGs is approximately 1 gene. These inconsistencies need to be corrected for accuracy.

Answer: We appreciate your attention to the details and improving our manuscript, which is crucial for ensuring the accuracy of our findings. In our initial analysis, we calculated the percentages based on the total number of genes that were significantly enriched within each GO term. This approach may have led to confusion and inconsistencies in our reporting. In response to your comment, we have thoroughly reviewed our calculations and made the necessary corrections. Specifically, we recalculated the number of genes associated with each GO term based on the total number of DEGs identified in our meta-analysis. We have ensured that these figures now align with the percentages presented in the text. We have updated both Figure 4 and the corresponding sections of the manuscript to reflect these corrections.

- The "PRISMA 2020 checklist" supplementary file contains important information about dataset adherence to MIAME guidelines, which is not addressed in the main manuscript. This information should be incorporated into the manuscript, as it is relevant to the quality of the data used.

Answer: Thank you for your valuable feedback regarding the inclusion of information from the "PRISMA 2020 checklist" supplementary file. In response, we have added detailed explanations in the "Methods" section, specifically the subsection "Data collecting". This addition highlights how our dataset complies with MIAME guidelines and emphasizes its relevance to the quality of the data used in our study.

Reviewer #2:

1) I think that the choice of research content is more meaningful. However, there are a lot of errors in the content, and it seems that the authors submitted this manuscript without checking it carefully, for example, there is an intersection of the Venn diagrams of the up- and down-regulated genes, and Fig2 in the manuscript actually refers to Fig3.

Answer: Thank you for your valuable feedback on our manuscript. We have carefully reviewed and revised the manuscript to address the errors you pointed out and similar items. Specifically, we have corrected the references to figures and meticulously checked to ensure clarity and adherence to grammatical standards. We believe these revisions enhance the quality of our manuscript, and we hope that it now meets the standards expected for publication.

2) In addition, the discussion section, although it is written a lot, it is not focused, and does not discuss the results of their own research, it is more like a review.

Answer: We appreciate your insights and have made significant revisions to enhance the focus and relevance of this section. In our revised version, we have shortened the discussion and ensured that it directly addresses the results of our own research. Additionally, we have removed unrelated materials and sources to streamline the content and improve clarity.

---

## [Decision Letter · Decision Letter 1]

17 Jan 2025

PONE-D-24-35105R1Comparative meta-analysis of barely transcriptome: pathogen type determines host preferencePLOS ONE

Dear Dr. Moghadam,

Thank you for submitting your manuscript to PLOS ONE. After careful consideration, we feel that it has merit but does not fully meet PLOS ONE’s publication criteria as it currently stands. Therefore, we invite you to submit a revised version of the manuscript that addresses the points raised during the review process.

We look forward to receiving your revised manuscript.

Kind regards,

Sumit Jangra, Ph.D.

Academic Editor

PLOS ONE

Journal Requirements:

Reviewers' comments:

Reviewer's Responses to Questions

**Comments to the Author**

1. If the authors have adequately addressed your comments raised in a previous round of review and you feel that this manuscript is now acceptable for publication, you may indicate that here to bypass the “Comments to the Author” section, enter your conflict of interest statement in the “Confidential to Editor” section, and submit your "Accept" recommendation.

Reviewer #3: All comments have been addressed

Reviewer #4: All comments have been addressed

Reviewer #5: All comments have been addressed

Reviewer #6: (No Response)

2. Is the manuscript technically sound, and do the data support the conclusions?

Reviewer #3: Yes

Reviewer #4: Yes

Reviewer #5: Yes

Reviewer #6: Yes

3. Has the statistical analysis been performed appropriately and rigorously? 

Reviewer #3: Yes

Reviewer #4: Yes

Reviewer #5: Yes

Reviewer #6: Yes

4. Have the authors made all data underlying the findings in their manuscript fully available?

Reviewer #3: Yes

Reviewer #4: Yes

Reviewer #5: Yes

Reviewer #6: Yes

5. Is the manuscript presented in an intelligible fashion and written in standard English?

Reviewer #3: Yes

Reviewer #4: Yes

Reviewer #5: Yes

Reviewer #6: Yes

6. Review Comments to the Author

Reviewer #3: Comments

-The experiment is interesting as explores genes involved in response of barley to pathogens attacks. As responses to biotic and abiotic stresses are sometime common in plant species, I recommend to read some interesting publications to address whether some of signaling and genes are common between the manuscript submitted and the following publications? Addressing such issues using the following citations may give better insight into better interpretation of the results in discussion or introduction parts of the manuscript.

Ravi S. et al. 2021. Development of an SNP Assay for Marker-Assisted Selection of Soil-Borne Rhizoctonia Solani AG-2-2-IIIB Resistance in Sugar Beet. Biology,11:49, //doi.org/10.3390/biology11010049

Shariatipur et al. 2021. Comparative Genomic Analysis of Quantitative Trait Loci Associated With Micronutrient Contents, Grain Quality, and Agronomic Traits in Wheat (Triticum aestivum L.). frontiers in Plant Science, //doi.org/10.3389/fpls.2021.709817

Shariatipur et al 2021. Meta-analysis of QTLome for grain zinc and iron contents in wheat (Triticum aestivum L.). Eyphytica 217, //doi.org/10.1007/s10681-021-02818-8

Salami et al. 2023. Integration of genome wide association studies (GWAS), metabolomics and transcriptomics reveals phenolic acids and flavonoids associated genes and their regulatory elements under drought stress in rapeseed flowers. Frontiers in Plant Science, 14, 10.3389/fpls.2023.1249142

Salami et al. 2024. Dissection of quantitative trait nucleotides and candidate genes associated with agronomic and yield-related traits under drought stress in rapeseed varieties: integration of genome-wide association study and transcriptomic analysis. Frontiers in Plant Sciences, 15 doi.org/10.3389/fpls.2024.1342359

Moazami et al. Optimization of agrobacterium mediated transformation of sugar beet: Glyphosate and insect pests resistance associated genes. Agronomy Journal, 112, //doi.org/10.1002/agj2.20384

-In Abstract, add number of data samples used for this meta- analysis

-Introduction is too long, I suggest to condense introduction and keep the most important sections related to the topic. I suggest merge some of similar paragraphs with same content

-Add to the table: the references where the data in Table 1 retrieved

-Some of publications suggested above analyzed WGCNA (almost Salami et al. papers). I suggest to have a look on these papers for interpretation of similar results for discussion part of the paper.

-Page 12, line 11, what does <<in following="" the="">> means? It is not needed as a start point in this line

-page 15 line 3, do not start a start sentence in a paragraphs with words such as , Next

-

Conclusion

-conclusion is too long and boring for readers. Conclusion should be in 3-4 sentences showing the most important finding. Si, revise the conclusion and focus on 3-4 main outcomes.</in>

Reviewer #4: General Comments:

Clarity and Conciseness:

The abstract contains technical terms and multiple ideas in each sentence, which may overwhelm the reader. Breaking down complex sentences into simpler ones would improve readability.

Structure and Flow:

While the abstract covers objectives, methods, results, and implications, the transitions between sections are abrupt. Adding linking phrases to guide the reader through the narrative would enhance flow.

Focus:

The abstract appears to attempt to cover too much ground, from meta-analysis to gene network analysis and signaling pathways. Focusing on a few key findings and their significance would increase its impact.

Specific Comments:

Opening Sentence:

"Fungi and insects have contrasting effects that alter the preference and performance of pathogenesis in barley."

This statement is too general and lacks context. Specify the nature of these "contrasting effects" to provide clarity. For instance, do fungi promote pathogenesis while insects inhibit it?

Objective Statement:

"Therefore, the characterization of synergistic and antagonistic pathogen mechanisms is highly important to be explored."

Revise for conciseness and clarity. Suggested rewrite:

"Characterizing synergistic and antagonistic pathogen mechanisms is crucial for understanding pathogenesis in barley."

Methods Section:

"We performed meta-analysis and co-expression gene network analysis of the barley transcriptome in response to fungi, aphids, and hormones."

Include a brief mention of the dataset size or sources to strengthen this statement's credibility.

Results Section:

"The results showed that 1.1% of DEGs were common between fungal and aphid-related datasets and 0.1% of DEGs were shared among all datasets."

Clarify the significance of these percentages. For instance, why are these common DEGs important?

"Promoter analysis revealed that AP2/EREBP and C2H2 zinc finger factors included the most frequent binding sites and played a more important role in connection with the SAR/ISR pathways."

Avoid vague terms like "more important role." Quantify or specify their role.

Hub Genes:

"Gene network analysis allocated the DEGs into multiple modules with high co-expression and identified specific pathogen-responsive hub genes, including SSI2, PAD2, RPS1, RPS17, SHM1, CYP5, and RPL21C."

Consider separating these findings into a new sentence to avoid overloading the reader. Mention the biological relevance of these hub genes briefly.

Novel Genes:

"Moreover, novel hub genes with unknown functions were identified, including AT4G31530, AT5G54855, AT3G07640, and AT1G05720, which may influence host preference against biotrophic/hemibiotrophic pathogens and aphids."

Highlight the potential significance of these novel genes more explicitly. How might they be used in future research or applications?

Conclusion:

"This is the first preliminary systems biology analysis of barley transcriptomic responses to heterotroph/biotroph cross-talk regarding the preference and performance of Rhopalosiphum padi, introducing valuable candidate genes that may be beneficial for accelerating genetic engineering programs."

This conclusion is lengthy and could be split into two sentences for clarity. Additionally, avoid terms like "preliminary" in an abstract unless necessary, as it may undermine the study's perceived value.

Suggestions for Improvement:

Use active voice where possible to make the abstract more engaging.

Add a sentence on the broader impact of this research, particularly for barley disease resistance or agricultural applications.

Ensure all technical terms (e.g., DEGs, SAR, ISR) are either commonly understood in the field or briefly explained.

Reviewer #5: (No Response)

Reviewer #6: 1.Clarification of trophic levels in the introduction

In the third paragraph of the introduction, only secondary consumers are exemplified, while the other two levels are mentioned generally. It would be helpful to provide specific examples for primary and tertiary consumers to ensure a balanced comparison.

2.Explanation of the variance threshold in meta-analysis

The manuscript states that 20% of DEGs with a variance of less than 0.2 were removed, but does not explain why 0.2 was chosen. The authors should justify this threshold.

3.Clarification of fold-change thresholds for up-regulated and down-regulated genes

The terms "up-regulated" and "down-regulated" are used, but no fold-change threshold is provided. The authors should specify criteria, such as "genes with a fold change of 2 or more are considered upregulated or downregulated."

4.Detailed description of DEG selection criteria

The manuscript lacks detailed information on the DEG selection process. The authors should clarify the criteria and statistical thresholds used.

7. PLOS authors have the option to publish the peer review history of their article (what does this mean? ). If published, this will include your full peer review and any attached files.

**Do you want your identity to be public for this peer review?** For information about this choice, including consent withdrawal, please see our Privacy Policy .

Reviewer #3: No

Reviewer #4: **Yes: ** Dr. Muraleedhar S Aski

Reviewer #5: No

Reviewer #6: No

---

## [Author Response · Author response to Decision Letter 2]

19 Feb 2025

Responses to the reviewers

Reviewer #3: Comments

1- The experiment is interesting as explores genes involved in response of barley to pathogens attacks. As responses to biotic and abiotic stresses are sometime common in plant species, I recommend to read some interesting publications to address whether some of signaling and genes are common between the manuscript submitted and the following publications? Addressing such issues using the following citations may give better insight into better interpretation of the results in discussion or introduction parts of the manuscript.

Ravi S. et al. 2021. Development of an SNP Assay for Marker-Assisted Selection of Soil-Borne Rhizoctonia Solani AG-2-2-IIIB Resistance in Sugar Beet. Biology,11:49, //doi.org/10.3390/biology11010049

Shariatipur et al. 2021. Comparative Genomic Analysis of Quantitative Trait Loci Associated With Micronutrient Contents, Grain Quality, and Agronomic Traits in Wheat (Triticum aestivum L.). frontiers in Plant Science, //doi.org/10.3389/fpls.2021.709817

Shariatipur et al 2021. Meta-analysis of QTLome for grain zinc and iron contents in wheat (Triticum aestivum L.). Eyphytica 217, //doi.org/10.1007/s10681-021-02818-8

Salami et al. 2023. Integration of genome wide association studies (GWAS), metabolomics and transcriptomics reveals phenolic acids and flavonoids associated genes and their regulatory elements under drought stress in rapeseed flowers. Frontiers in Plant Science, 14, 10.3389/fpls.2023.1249142

Salami et al. 2024. Dissection of quantitative trait nucleotides and candidate genes associated with agronomic and yield-related traits under drought stress in rapeseed varieties: integration of genome-wide association study and transcriptomic analysis. Frontiers in Plant Sciences, 15 doi.org/10.3389/fpls.2024.1342359

Moazami et al. Optimization of agrobacterium mediated transformation of sugar beet: Glyphosate and insect pests resistance associated genes. Agronomy Journal, 112, //doi.org/10.1002/agj2.20384

Answer: Thank you very much for your valuable suggestions about the references. We used these references and cited them in the introduction, results, and discussion.

2- In Abstract, add number of data samples used for this meta- analysis

Answer: Yes, we added.

3- Introduction is too long, I suggest to condense introduction and keep the most important sections related to the topic. I suggest merge some of similar paragraphs with same content

Answer: Thank you for your valuable feedback. We have merged similar paragraphs, removed redundant information, and focused on the most critical points to make it more concise and suitable. The structure is now more focused and avoids unnecessary repetition.

4- Add to the table: the references where the data in Table 1 retrieved

Answer: Yes, we added

5- Some of publications suggested above analyzed WGCNA (almost Salami et al. papers). I suggest to have a look on these papers for interpretation of similar results for discussion part of the paper.

Answer: Thank you very much. For sure, it was very useful. We used it and cited it.

6- Page 12, line 11, what does <> means? It is not needed as a start point in this line

Answer: We removed it.

7- Page 15 line 3, do not start a start sentence in a paragraphs with words such as, Next

Answer: Yes, we replaced them with more appropriate words.

Conclusion

8- conclusion is too long and boring for readers. Conclusion should be in 3-4 sentences showing the most important finding. Si, revise the conclusion and focus on 3-4 main outcomes.

Answer: We edited it based on your comment.

Reviewer #4:

General Comments:

Clarity and Conciseness:

9- The abstract contains technical terms and multiple ideas in each sentence, which may overwhelm the reader. Breaking down complex sentences into simpler ones would improve readability.

Answer: Yes, it was improved based on your comment.

Structure and Flow:

10- While the abstract covers objectives, methods, results, and implications, the transitions between sections are abrupt. Adding linking phrases to guide the reader through the narrative would enhance flow.

Answer: Yes, it was improved.

Focus:

11- The abstract appears to attempt to cover too much ground, from meta-analysis to gene network analysis and signaling pathways. Focusing on a few key findings and their significance would increase its impact.

Answer: Yes, it was improved.

Specific Comments:

Opening Sentence:

12- "Fungi and insects have contrasting effects that alter the preference and performance of pathogenesis in barley."

This statement is too general and lacks context. Specify the nature of these "contrasting effects" to provide clarity. For instance, do fungi promote pathogenesis while insects inhibit it?

Answer: We thank the reviewer for their comment. To address the concern, we have revised the opening sentence to specify and clarify the contrasting effects of fungi and insects on pathogenesis in barley.

Objective Statement:

13- "Therefore, the characterization of synergistic and antagonistic pathogen mechanisms is highly important to be explored."

Revise for conciseness and clarity. Suggested rewrite:

"Characterizing synergistic and antagonistic pathogen mechanisms is crucial for understanding pathogenesis in barley."

Answer: Thank you for your valuable feedback and constructive suggestion. We have revised the sentence as recommended to improve conciseness and clarity.

Methods Section:

14- "We performed meta-analysis and co-expression gene network analysis of the barley transcriptome in response to fungi, aphids, and hormones."

Include a brief mention of the dataset size or sources to strengthen this statement's credibility.

Answer: Yes, it was applied.

Results Section:

15- "The results showed that 1.1% of DEGs were common between fungal and aphid-related datasets and 0.1% of DEGs were shared among all datasets."

Clarify the significance of these percentages. For instance, why are these common DEGs important?

Answer: Thank you for your valuable feedback. We clarified these parts based on your comment.

16- "Promoter analysis revealed that AP2/EREBP and C2H2 zinc finger factors included the most frequent binding sites and played a more important role in connection with the SAR/ISR pathways."

Avoid vague terms like "more important role." Quantify or specify their role.

Answer: Thank you for your valuable feedback. We specified this part and other similar results.

Hub Genes:

17- "Gene network analysis allocated the DEGs into multiple modules with high co-expression and identified specific pathogen-responsive hub genes, including SSI2, PAD2, RPS1, RPS17, SHM1, CYP5, and RPL21C."

Consider separating these findings into a new sentence to avoid overloading the reader. Mention the biological relevance of these hub genes briefly.

Answer: Thank you for your valuable feedback. We improved the descriptions based on your comment.

Novel Genes:

18- "Moreover, novel hub genes with unknown functions were identified, including AT4G31530, AT5G54855, AT3G07640, and AT1G05720, which may influence host preference against biotrophic/hemibiotrophic pathogens and aphids."

Highlight the potential significance of these novel genes more explicitly. How might they be used in future research or applications?

Answer: Thank you for your valuable feedback. We clarified these parts based on your comment.

Conclusion:

19- "This is the first preliminary systems biology analysis of barley transcriptomic responses to heterotroph/biotroph cross-talk regarding the preference and performance of Rhopalosiphum padi, introducing valuable candidate genes that may be beneficial for accelerating genetic engineering programs."

This conclusion is lengthy and could be split into two sentences for clarity. Additionally, avoid terms like "preliminary" in an abstract unless necessary, as it may undermine the study's perceived value.

Answer: Thank you. We edited this sentence based on your comment.

Suggestions for Improvement:

20- Use active voice where possible to make the abstract more engaging.

Answer: Thank you for your comment. It was improved.

21- Add a sentence on the broader impact of this research, particularly for barley disease resistance or agricultural applications.

Answer: Yes, it was added.

22- Ensure all technical terms (e.g., DEGs, SAR, ISR) are either commonly understood in the field or briefly explained.

Answer: Thanks, we would like to clarify that all abbreviations terms, including DEGs (Differentially Expressed Genes), SAR (Systemic Acquired Resistance), and ISR (Induced Systemic Resistance), were fully defined upon their first use and abbreviations section. As suggested, we have added brief explanations for the terms SAR, ISR, and DEGs in the relevant paragraphs of the manuscript. These additions ensure that readers unfamiliar with these terms can easily understand their significance in the context of our study.

Reviewer #5: (No Response)

Reviewer #6:

23. Clarification of trophic levels in the introduction

In the third paragraph of the introduction, only secondary consumers are exemplified, while the other two levels are mentioned generally. It would be helpful to provide specific examples for primary and tertiary consumers to ensure a balanced comparison.

Answer: Thank you for your valuable comment. We have revised the introduction to include specific examples of primary/secondary/tertiary consumers to provide a more balanced and comprehensive discussion of trophic levels in the context of host-pathogen interactions.

24- Explanation of the variance threshold in meta-analysis

The manuscript states that 20% of DEGs with a variance of less than 0.2 were removed, but does not explain why 0.2 was chosen. The authors should justify this threshold.

Answer: Thank you for your valuable feedback. The choice of a variance threshold of 0.2 was based on statistical considerations to ensure the robustness and reliability of the DEGs identified in our study.

The MetaDE package, which implements the RankProd method, is designed to integrate multiple datasets and identify DEGs with consistent expression patterns across studies. The RankProd method is non-parametric and relies on rank-based statistics, making it robust to outliers and noise. Specifically, variance thresholds are commonly used to filter out genes with low variability, as these genes often contribute noise rather than meaningful biological signals. The increasing the variance > 0.2 led to the exclusion of key genes that have a lot of biological relevance. However, genes with very low variance (e.g., variance < 0.2) are often non-informative or exhibit minimal biological relevance, as they show little to no change in expression across conditions. Including such genes can increase the risk of false positives and unnecessary computational burden.

By removing the bottom 20% of DEGs with a variance of 0.2, we aimed to remove genes with low variety in different conditions or treatments and focus on genes with significant expression changes, resulting in improved statistical power and biological relevance of our meta-analysis. In meta-analysis, managing the large volume of data and reducing its complexity is very important. By eliminating genes with low variance, analyses become simpler and more understandable. This threshold was chosen based on empirical evidence from previous studies using similar approaches (for example, [Soltani et al. (2023), Ramasamy et al. (2008) and Hong et al. (2006]), where a variance cutoff of 0.2 effectively balanced the trade-off between retaining biologically relevant genes and reducing noise. We believe that using this threshold will increase the accuracy and validity of our meta-analysis results. We added a brief explanation to the Methods section of the manuscript to clarify this point.

25. Clarification of fold-change thresholds for up-regulated and down-regulated genes

The terms "up-regulated" and "down-regulated" are used, but no fold-change threshold is provided. The authors should specify criteria, such as "genes with a fold change of 2 or more are considered upregulated or downregulated."

Answer: Yes, we improved these results based on your comment.

26. Detailed description of DEG selection criteria

The manuscript lacks detailed information on the DEG selection process. The authors should clarify the criteria and statistical thresholds used.

Answer: We clarify that our selection criteria and statistical thresholds for DEGs included a log2 fold change (FC) greater than 1 or less than -1, coupled with a false discovery rate (FDR) of less than or equal to 0.05."

---

## [Decision Letter · Decision Letter 2]

24 Feb 2025

Comparative meta-analysis of barely transcriptome: pathogen type determines host preference

PONE-D-24-35105R2

Dear Dr. Moghadam,

We’re pleased to inform you that your manuscript has been judged scientifically suitable for publication and will be formally accepted for publication once it meets all outstanding technical requirements.

Kind regards,

Sumit Jangra, Ph.D.

Academic Editor

PLOS ONE

Additional Editor Comments (optional):

Reviewers' comments:

Reviewer's Responses to Questions

**Comments to the Author**

1. If the authors have adequately addressed your comments raised in a previous round of review and you feel that this manuscript is now acceptable for publication, you may indicate that here to bypass the “Comments to the Author” section, enter your conflict of interest statement in the “Confidential to Editor” section, and submit your "Accept" recommendation.

Reviewer #3: All comments have been addressed

2. Is the manuscript technically sound, and do the data support the conclusions?

Reviewer #3: Yes

3. Has the statistical analysis been performed appropriately and rigorously? 

Reviewer #3: Yes

4. Have the authors made all data underlying the findings in their manuscript fully available?

Reviewer #3: Yes

5. Is the manuscript presented in an intelligible fashion and written in standard English?

Reviewer #3: Yes

6. Review Comments to the Author

Reviewer #3: All comments addressed properly. I suggest accept for this version ans it is adequate for publication

7. PLOS authors have the option to publish the peer review history of their article (what does this mean? ). If published, this will include your full peer review and any attached files.

**Do you want your identity to be public for this peer review?** For information about this choice, including consent withdrawal, please see our Privacy Policy .

Reviewer #3: No

---

## [Editor Report · Acceptance letter]

PONE-D-24-35105R2

PLOS ONE

Dear Dr. Moghadam,

I'm pleased to inform you that your manuscript has been deemed suitable for publication in PLOS ONE. Congratulations! Your manuscript is now being handed over to our production team.

Kind regards,

on behalf of

Dr. Sumit Jangra

Academic Editor

PLOS ONE